# LKB1 drives stasis and C/EBP-mediated reprogramming to an alveolar type II fate in lung cancer

Christopher W. Murray[1], Jennifer J. Brady[2], Mingqi Han[3,4], Hongchen Cai[2], Min K. Tsai [1], Sarah E. Pierce [1], Ran Cheng [5,6], Janos Demeter [5], David M. Feldser [7,8], Peter K. Jackson [1,5,9,10], David B. Shackelford [3,4] & Monte M. Winslow [1,2,9,10 ✉]

*LKB1* is among the most frequently altered tumor suppressors in lung adenocarcinoma. Inactivation of *Lkb1* accelerates the growth and progression of oncogenic KRAS-driven lung tumors in mouse models. However, the molecular mechanisms by which LKB1 constrains lung tumorigenesis and whether the cancer state that stems from *Lkb1* deficiency can be reverted remains unknown. To identify the processes governed by LKB1 in vivo, we generated an allele which enables *Lkb1* inactivation at tumor initiation and subsequent *Lkb1* restoration in established tumors. Restoration of *Lkb1* in oncogenic KRAS-driven lung tumors suppressed proliferation and led to tumor stasis. *Lkb1* restoration activated targets of C/EBP transcription factors and drove neoplastic cells from a progenitor-like state to a less proliferative alveolar type II cell-like state. We show that C/EBP transcription factors govern a subset of genes that are induced by LKB1 and depend upon NKX2-1. We also demonstrate that a defining factor of the alveolar type II lineage, C/EBPα, constrains oncogenic KRAS-driven lung tumor growth in vivo. Thus, this key tumor suppressor regulates lineage-specific transcription factors, thereby constraining lung tumor development through enforced differentiation.

[1] Cancer Biology Program, Stanford University School of Medicine, Stanford, CA, USA. [2] Department of Genetics, Stanford University School of Medicine, Stanford, CA, USA. [3] Division of Pulmonary and Critical Care Medicine, Department of Medicine, David Geffen School of Medicine, University of California, Los Angeles, CA, USA. [4] Jonsson Comprehensive Cancer Center, David Geffen School of Medicine, University of California, Los Angeles, CA, USA. [5] Baxter Laboratory, Department of Microbiology & Immunology, Stanford University School of Medicine, Stanford, CA 94305, USA. [6] Department of Biology, Stanford University, Stanford, CA 94305, USA. [7] Department of Cancer Biology, University of Pennsylvania, Philadelphia, PA 19104-6160, USA. [8] Abramson Family Cancer Research Institute, University of Pennsylvania, Philadelphia, PA 19104-6160, USA. [9] Department of Pathology, Stanford University School of Medicine, Stanford, CA, USA. [10] Stanford Cancer Institute, Stanford University School of Medicine, Stanford, CA, USA. ✉email: mwinslow@stanford.edu

Neoplastic cells undergo a series of cell-state transitions throughout cancer development[1–5]. Genetic alterations, including the inactivation of tumor suppressor genes, allow cells to bypass key checkpoints that constrain the transition from normal to malignant states[6]. While genomic analyses of human cancers have uncovered a multitude of putative tumor suppressor genes, many of these genes have yet to be fully characterized with respect to the molecular and cellular processes that they govern to suppress tumorigenesis, including the maintenance of terminal differentiation[6–10]. The ability to globally or conditionally inactivate genes in mice has served as the basis for studying tumor suppressor function in vivo for almost three decades[11–14]. However, beyond demonstrating tumor-suppressive capacity and providing tumor models through which genotype-specific cellular and molecular features can be uncovered, knockout models provide limited information regarding the direct mechanisms by which tumor suppressors block tumor formation and progression[15].

Advances in conditional and inducible gene regulation have laid the foundation for the development of reversible genetic systems in which tumor suppressors can be inactivated and subsequently restored within established tumors in vivo[15]. By coupling these strategies with unbiased cellular and molecular profiling, it is possible to not only examine the consequences of tumor suppressor loss but also identify latent programs that are re-initiated upon tumor suppressor reactivation, some of which are likely critical for tumor suppression[15]. In addition to guiding mechanistic interrogation, in vivo tumor suppressor restoration approaches have the potential to uncover the extent to which the maintenance of a neoplastic state depends upon the sustained inactivation of a given tumor suppressor. While the dependence on oncogene activity is well-established and supported by the clinical success of oncogene-targeted therapies, the consequences of reactivating tumor suppressors are much less understood[16,17]. Fascinatingly, studies on some of the most frequently inactivated tumor suppressors have revealed that the restoration of different tumor suppressors in vivo drives distinct phenotypic outcomes (from complete regression after *Apc* restoration to inhibition of metastatic progression after *Rb1* restoration)[18–22]. Thus, restoration approaches uniquely establish causal links between tumor suppressors and the processes that they govern, as well as reveal the various manners in which tumors respond to their reactivation.

The tumor suppressor LKB1 (also known as serine/threonine kinase 11; *STK11*) is frequently inactivated in several human cancer types and governs differentiation in both normal and neoplastic settings[23,24]. In lung adenocarcinoma, *LKB1* is genetically disrupted in 15–30% of tumors, and its deletion in mouse models of lung cancer dramatically accelerates lung tumor growth[25,26]. Through the manipulation of LKB1 in lung adenocarcinoma cell lines in vitro and comparative analyses between *LKB1* wild-type and mutant lung tumors, many consequences of *LKB1* inactivation have been identified, including oxidative and ER stress, unique metabolic dependencies and therapeutic vulnerabilities, as well as an immunosuppressive microenvironment[27–34]. Recent work has uncovered a role for the salt-inducible kinases (SIKs), particularly SIK1 and SIK3, as immediate downstream effectors that are critical for LKB1-mediated lung tumor suppression[35,36]. However, our understanding of the molecular effectors downstream of the LKB1-SIK axis that are critical for tumor suppression in vivo remains limited. Furthermore, whether the highly aggressive state that emerges as a consequence of *Lkb1* deficiency can be reverted remains an outstanding question with implications for the value of therapeutic strategies to counteract specific features of the *Lkb1*-deficient state[27,29,31,37,38].

Here, we show that restoration of *Lkb1* in lung tumors in vivo suppresses proliferation and induces tumor stasis. Through unbiased transcriptomic and proteomic profiling of the response to *Lkb1* restoration, we uncover a requirement for LKB1 in the maintenance of alveolar type II cell identity, as well as define a connection between LKB1 activity and the induction of C/EBP target genes that are co-regulated by NKX2-1. We also demonstrate that a defining factor of the alveolar type II lineage, C/EBPα, suppresses tumor growth. Thus, we establish a link between tumor suppression in lung cancer and the activity of lineage-defining transcription factors, the disruption of which results in reversion to a progenitor-like state.

## Results

### Generation of a conditionally inactivatable and restorable *Lkb1$^{XTR}$* allele.

To investigate the cellular and molecular processes governed by LKB1 in vivo, we generated an *Lkb1$^{XTR}$* allele with which we could conditionally inactivate and subsequently restore *Lkb1* within autochthonous tumors (Fig. 1a and "Methods")[19]. We inserted an inverted gene trap cassette flanked by heterotypic pairs of mutant *loxP* sites and nested between FRT sites within the first intron of *Lkb1*. This design enables Cre-mediated stable inversion of the gene trap to intercept endogenous splicing and subsequent FLPo-ER$^{T2}$-mediated deletion of the cassette upon tamoxifen administration (Fig. 1a and Supplementary Fig. 1a–d). Despite reduced levels of *Lkb1* mRNA and protein in various tissues from *Lkb1$^{XTR/XTR}$* mice as compared to *Lkb1* wild-type mice, *Lkb1$^{XTR/XTR}$* mice developed normally, were born at the expected Mendelian ratio, and did not develop gastrointestinal polyps as would be expected if the *Lkb1$^{XTR}$* allele greatly compromised LKB1 tumor suppressor activity (Supplementary Fig. 2a–d)[39]. Consistent with homozygous inactivation of *Lkb1* leading to embryonic lethality, we were unable to obtain *Lkb1$^{TR/TR}$* mice upon intercrossing *Lkb1$^{TR/+}$* mice (generated by crossing *Lkb1$^{XTR}$* mice to a Cre deleter line; Supplementary Fig. 2e)[40,41]. Furthermore, *Lkb1$^{TR/+}$* mice developed gastrointestinal polyps that were histologically similar to those in *Lkb1$^{null/+}$* mice (Supplementary Fig. 2d)[39]. Together, these findings demonstrate that the *Lkb1$^{XTR}$* allele, in the expressed conformation, retains tumor-suppressive function, while the trapped *Lkb1$^{TR}$* conformation disrupts *Lkb1* expression.

### *Lkb1* inactivation with the *Lkb1$^{XTR}$* allele increases lung tumor burden and *Lkb1* restoration dramatically decreases lung tumor burden.

Next, we crossed the *Lkb1$^{XTR}$* allele into the *Kras$^{LSL-G12D/+}$;Rosa26$^{LSL-tdTomato}$* (*KT*) background to generate *KT;Lkb1$^{XTR/XTR}$* mice for the initiation of oncogenic KRAS-driven lung tumors after intratracheal delivery of lentiviral Cre (Supplementary Fig. 2f). Consistent with previous results using an *Lkb1$^{flox}$* allele or CRISPR/Cas9-mediated targeting, gene trap-mediated inactivation of *Lkb1* in *KT;Lkb1$^{XTR/XTR}$* mice dramatically increased lung tumor burden relative to *KT* mice (Supplementary Fig. 2g–k)[25,42]. As anticipated, lung tumors in *KT;Lkb1$^{XTR/XTR}$* mice were adenomas and adenocarcinomas that express NKX2-1, a marker of adenocarcinoma differentiation, with only rare clusters of poorly differentiated cancer cells at late time points (Supplementary Fig. 2l, m)[36,43,44]. Together, these data indicate that the *Lkb1$^{XTR}$* allele operates as designed to disrupt *Lkb1*.

To examine the impact of *Lkb1* restoration on tumor growth, we generated *KT;Lkb1$^{XTR/XTR}$* mice with the *Rosa26$^{FLPo-ERT2}$* (FLPo-ER$^{T2}$) allele such that *Lkb1* expression could be restored within established, *Lkb1*-deficient tumors upon treatment with tamoxifen (Fig. 1b). We initiated tumors in *KT, KT;Lkb1$^{XTR/XTR}$* and

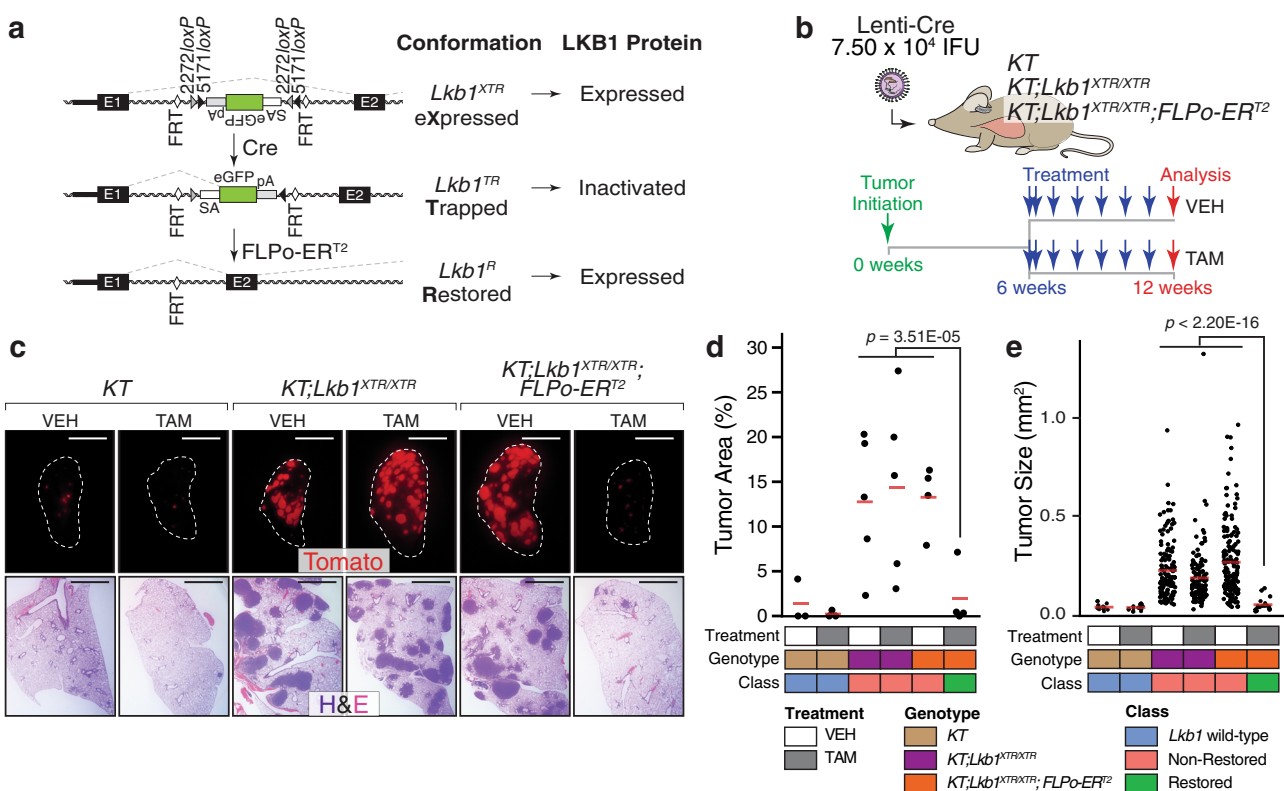

**Fig. 1 Lkb1 restoration in established lung tumors dramatically decreases lung tumor burden. a** Schematic of the XTR cassette inserted within the first intron of *Lkb1* in the eXpressed, Trapped, and Restored conformations. The XTR cassette is composed of an inverted gene trap consisting of an Ad40 splice acceptor upstream of eGFP. The gene trap is flanked by heterotypic *loxP* sites to allow for stable Cre-mediated inversion, which results in the truncation of wild-type *Lkb1* transcripts. The gene trap is nested between two FRT sites to enable FLPo-ER$^{T2}$-mediated deletion in the presence of tamoxifen, which results in the restoration of wild-type *Lkb1* transcripts and LKB1 protein. **b** Assessment of the impact of *Lkb1* restoration on tumor burden. Lung tumors were initiated in *KT, KT;Lkb1$^{XTR/XTR}$*, and *KT;Lkb1$^{XTR/XTR}$;FLPo-ER$^{T2}$* mice. At 6 weeks post-initiation, tumor-bearing mice were treated for 6 weeks with corn oil vehicle or tamoxifen prior to analysis. IFU infectious units, VEH vehicle, TAM tamoxifen. **c** Representative fluorescence (top) and hematoxylin–eosin (H&E) staining (bottom) images of tumor-bearing lungs from *KT, KT;Lkb1$^{XTR/XTR}$*, and *KT;Lkb1$^{XTR/XTR}$;FLPo-ER$^{T2}$* mice treated with vehicle or tamoxifen. Lung lobes within fluorescent images are outlined in white. Top scale bars = 5 mm. Bottom scale bars = 2 mm. **d, e** Tumor area (**d**) and tumor size (**e**) as assessed by histology for tumor-bearing *KT, KT;Lkb1$^{XTR/XTR}$*, and *KT;Lkb1$^{XTR/XTR}$;FLPo-ER$^{T2}$* mice at 12 weeks after tumor initiation, following 6 weeks of treatment with vehicle or tamoxifen. In **d**, each dot represents a mouse, while each dot in **e** corresponds to a tumor. Red crossbars indicate the mean. In **d**, *KT*-vehicle, n = 3 mice; *KT*-tamoxifen, n = 3 mice; *KT;Lkb1$^{XTR/XTR}$*-vehicle, n = 5 mice; *KT;Lkb1$^{XTR/XTR}$*-tamoxifen, n = 5 mice; *KT;Lkb1$^{XTR/XTR}$;FLPo-ER$^{T2}$*-vehicle, n = 4 mice; *KT;Lkb1$^{XTR/XTR}$;FLPo-ER$^{T2}$*-tamoxifen, n = 4 mice. In **e**, *KT*-vehicle, n = 11 tumors; *KT*-tamoxifen, n = 14 tumors; *KT;Lkb1$^{XTR/XTR}$*-vehicle, n = 147 tumors; *KT;Lkb1$^{XTR/XTR}$*-tamoxifen, n = 135 tumors; *KT;Lkb1$^{XTR/XTR}$;FLPo-ER$^{T2}$*-vehicle, n = 167 tumors; *KT;Lkb1$^{XTR/XTR}$;FLPo-ER$^{T2}$*-tamoxifen, n = 19 tumors. One tissue section per mouse was analyzed. *P* values calculated by two-sided unpaired *t* test. VEH vehicle, TAM tamoxifen.

*KT;Lkb1$^{XTR/XTR}$;FLPo-ER$^{T2}$* mice with lentiviral Cre and began weekly administration of either corn oil vehicle or tamoxifen at 6 weeks after tumor initiation. Six weeks after *Lkb1* restoration, tumor burden was markedly decreased in the restored context, including sevenfold fewer surface tumors, sixfold reduced total tumor area, and fourfold decreased average tumor size (Fig. 1c–e and Supplementary Fig. 3a, b). Strikingly, the tumor burden in restored mice was comparable to that of *KT* mice, suggesting that *Lkb1* restoration at early stages of tumorigenesis dramatically impairs tumor growth. Tamoxifen treatment of *KT;Lkb1$^{XTR/XTR}$* mice (which lack FLPo-ER$^{T2}$) had no impact on tumor burden (Fig. 1c–e and Supplementary Fig. 3a, b). *Lkb1* restoration at 6 weeks after tumor initiation almost doubled median survival (from 18 to 32 weeks), thus underscoring the dramatic impact of *Lkb1* restoration on tumor growth (Supplementary Fig. 3c, d). Wild-type LKB1 protein was undetectable in neoplastic cells from non-restored tumors and comparable to that of *Lkb1* wild-type tumors after restoration (Supplementary Fig. 3e). These findings demonstrate that established tumors remain susceptible to the tumor-suppressive activity of LKB1.

To further assess the impact of *Lkb1* restoration, we transplanted neoplastic cells from tumors in *KT;Lkb1$^{XTR/XTR}$;FLPo-ER$^{T2}$* donor mice into recipient mice via intratracheal delivery (Supplementary Fig. 4a). Following a 3-week period of engraftment, recipient mice were either analyzed, treated with vehicle, or treated with tamoxifen. Analysis after an additional 5 weeks indicated that *Lkb1* restoration decreased the number of tdTomato$^{positive}$ surface tumors by fourfold and reduced the total tumor area by 15–25-fold relative to vehicle treatment (Supplementary Fig. 4b–d). Surprisingly, the tumor burden after five weeks of *Lkb1* restoration was comparable to that of recipient mice analyzed after only the initial 3 weeks of growth (Supplementary Fig. 4b–d). Furthermore, in a separate experiment, *Lkb1* restoration prior to transplantation dramatically decreased the number of tdTomato$^{positive}$ surface tumors, suggesting that *Lkb1* restoration might also reduce the fraction of tumor-engrafting cells (Supplementary Fig. 4e, f). These observations underscore a critical role for LKB1 in suppressing multiple aspects of lung tumor growth and tumor-engrafting capacity.

Given the critical role of the p53 tumor suppressor in lung adenocarcinoma, as well as the functional link between LKB1 and the pro-apoptotic and growth-suppressive functions of p53, we determined whether concomitant inactivation of *Trp53* would abrogate the growth-suppressive effects of *Lkb1* restoration[45–49]. To assess the effects of restoration in the absence of p53, we initiated lung tumors in *KT;Trp53*$^{flox/flox}$ (*KPT*);*Lkb1*$^{XTR/XTR}$ and *KPT;Lkb1*$^{XTR/XTR}$;*FLPo-ER*$^{T2}$ mice and began vehicle or tamoxifen treatment at 6 weeks after tumor initiation (Supplementary Fig. 5a). After 6 weeks of *Lkb1* restoration, tumor burden was significantly decreased, albeit to a lesser extent as compared to the *Trp53* wild-type setting, including a twofold decrease in total lung weight and total tumor area, and a nearly fourfold reduction in average tumor size (Supplementary Fig. 5b–f, Fig. 1c–e, and Supplementary Fig. 3a, b). The reduced impact of *Lkb1* restoration in the context of *Trp53*-deficiency could result from more rapid tumor growth, progression prior to restoration, and/or a partial requirement for p53 in LKB1-driven growth arrest.

**Lkb1 restoration impairs lung tumor growth and decreases glucose avidity.** Given the dramatic effect of *Lkb1* restoration on lung tumor burden, we examined the impact of *Lkb1* restoration on the dynamics of lung tumor growth by performing longitudinal micro-computed tomography (μCT) imaging on restored and non-restored tumors. We initiated tumors in *KT;Lkb1*$^{XTR/XTR}$ and *KT;Lkb1*$^{XTR/XTR}$;*FLPo-ER*$^{T2}$ mice with lentiviral Cre, began weekly treatment with either vehicle or tamoxifen upon detection of lung nodules (which ranged from 17 to 21 weeks after tumor initiation), and tracked tumor volume by μCT for 6–10 weeks (Fig. 2a and Supplementary Fig. 6a). While non-restored tumors continued to grow, restored tumors were arrested (Fig. 2b, c and Supplementary Fig. 6b). Histological examination revealed that *Lkb1* restoration greatly reduced tumor burden, including a sevenfold decrease in total tumor area and a fivefold reduction in individual tumor size (Supplementary Fig. 6c, d). These data demonstrate that the restoration of *Lkb1*, unlike other tumor suppressors, results in profound tumor stasis without regression.

To determine at the cellular level how *Lkb1* restoration drives tumor stasis, we examined markers of proliferation (BrdU incorporation and Ki-67) and cell death (cleaved caspase 3) by immunohistochemistry 2 weeks following *Lkb1* restoration (Supplementary Fig. 7a). *Lkb1*-restored tumors were significantly less proliferative as compared to non-restored tumors, without evidence of increased cell death (Fig. 2d–f and Supplementary Fig. 7b, c). Consistently, *Lkb1* restoration in lung cancer cell lines derived from tumors from *KPT;Lkb1*$^{XTR/XTR}$;*FLPo-ER*$^{T2}$ mice resulted in a significant decrease in the fraction of cells in S phase and variable effects on the rate of cell death after *Lkb1* restoration (Supplementary Fig. 7d–g). Thus, the induction of tumor stasis by *Lkb1* restoration is likely driven by suppression of proliferation.

*Lkb1* loss has been previously linked to enhanced glucose uptake in mouse and human lung tumors as well as an increased glycolytic flux in human lung cancer cells in vitro[50,51]. To monitor changes in glucose uptake in response to *Lkb1* restoration, we performed serial positron emission tomography with 2-deoxy-2-[fluorine-18]fluoro-D-glucose integrated with computed tomography ($^{18}$F-FDG-PET/CT) imaging (Fig. 2g and Supplementary Fig. 8a). Tumors were initiated in *KT;Lkb1*$^{XTR/XTR}$ and *KT;Lkb1*$^{XTR/XTR}$;*FLPo-ER*$^{T2}$ mice with lentiviral Cre, and mice were treated with tamoxifen after establishing a baseline of $^{18}$F-FDG uptake (two consecutive measurements of $^{18}$F-FDG uptake). Within 2 weeks of starting tamoxifen treatment, restored tumors had reduced uptake relative to pre-treatment levels, while non-restored tumors trended

towards increased $^{18}$F-FDG uptake (Fig. 2h and Supplementary Fig. 8b, c). Six weeks after treatment initiation, $^{18}$F-FDG uptake had increased nearly twofold relative to pre-treatment among non-restored tumors, whereas it remained largely unchanged in the *Lkb1*-restored context. Even 12 weeks after *Lkb1* restoration, $^{18}$F-FDG uptake remained unchanged (Fig. 2h and Supplementary Fig. 8c). These data demonstrate that *Lkb1* restoration induces tumor stasis and abrogates the increase in glucose avidity that coincides with tumor progression.

**Lkb1 restoration drives transcriptional programs relating to alveolar type II cell functions.** To uncover the molecular processes governed by LKB1 in vivo, we performed RNA-seq on neoplastic cells that were isolated by FACS from restored and non-restored lung tumors 2 weeks after starting tamoxifen treatment, as well as *Lkb1* wild-type tumors from *KT* mice (Fig. 3a and Supplementary Data 1–4). The expression of *Lkb1* was negligible in non-restored tumors, but comparable in restored and *Lkb1* wild-type tumors (Supplementary Fig. 9a). Comparison of non-restored to *KT* tumors revealed that the gene expression differences were similar to published comparisons of *Lkb1*-deficient and *Lkb1*-proficient mouse lung tumors (Supplementary Fig. 9b). Furthermore, cancer cells from non-restored tumors had many established transcriptional features of the *Lkb1*-deficient state, including higher expression of gene sets relating to angiogenesis, hypoxia, adhesion, and epithelial-mesenchymal transition (Supplementary Fig. 9c)[25,35,36,43,52,53].

To understand how the *Lkb1*-restored state relates to the *KT* and non-restored states, we performed hierarchical clustering and principal component analysis. Hierarchical clustering revealed that restored tumors co-segregated with *KT* tumors away from non-restored tumors (Fig. 3b). By principal component analysis, *Lkb1* wild-type, non-restored, and restored tumors were separated across the first principal component (Supplementary Fig. 9d). *Lkb1*-restored tumors clustered at an intermediate position between *KT* and non-restored tumors, indicating that the acute response to *Lkb1* restoration results in partial reversion to a transcriptional state that remains distinct from tumors that were *Lkb1*-proficient throughout their development (Supplementary Fig. 9d). Genes with the lowest loading coefficients with respect to the first principal component (i.e. genes that are highest in *KT* tumors relative to restored and non-restored tumors) were enriched for gene sets relating to chemotaxis, extracellular structure organization, protein secretion, and steroid metabolism (Supplementary Fig. 9e). These programs are reminiscent of the surfactant production and immunomodulatory functions of ATII cells, which are thought to be a major cell type of origin for oncogenic KRAS-driven lung adenocarcinoma[54–56]. In contrast, the genes with the highest loading coefficients within the first principal component were enriched for gene sets relating to proliferation, adhesion, and extracellular matrix interactions, which are processes that have been linked to early progenitors of the distal lung epithelium (Supplementary Fig. 9e)[54]. These observations suggest that LKB1 activity may govern a transition between cycling progenitor-like and non-cycling ATII-like states.

Next, we performed *k*-means clustering to identify sets of genes that change concordantly across all samples (Fig. 3c). Genes that were higher in both restored and *Lkb1* wild-type tumors were enriched for gene sets related to antigen presentation and lipid metabolism, which are again consistent with established ATII cell functions (Supplementary Fig. 10a)[57,58]. Genes relating to angiogenesis and adhesion were specifically higher in non-restored tumors (Supplementary Fig. 10a). Apart from these changes at the global level, direct comparison of restored and non-restored tumors suggested that LKB1 reduces proliferation-

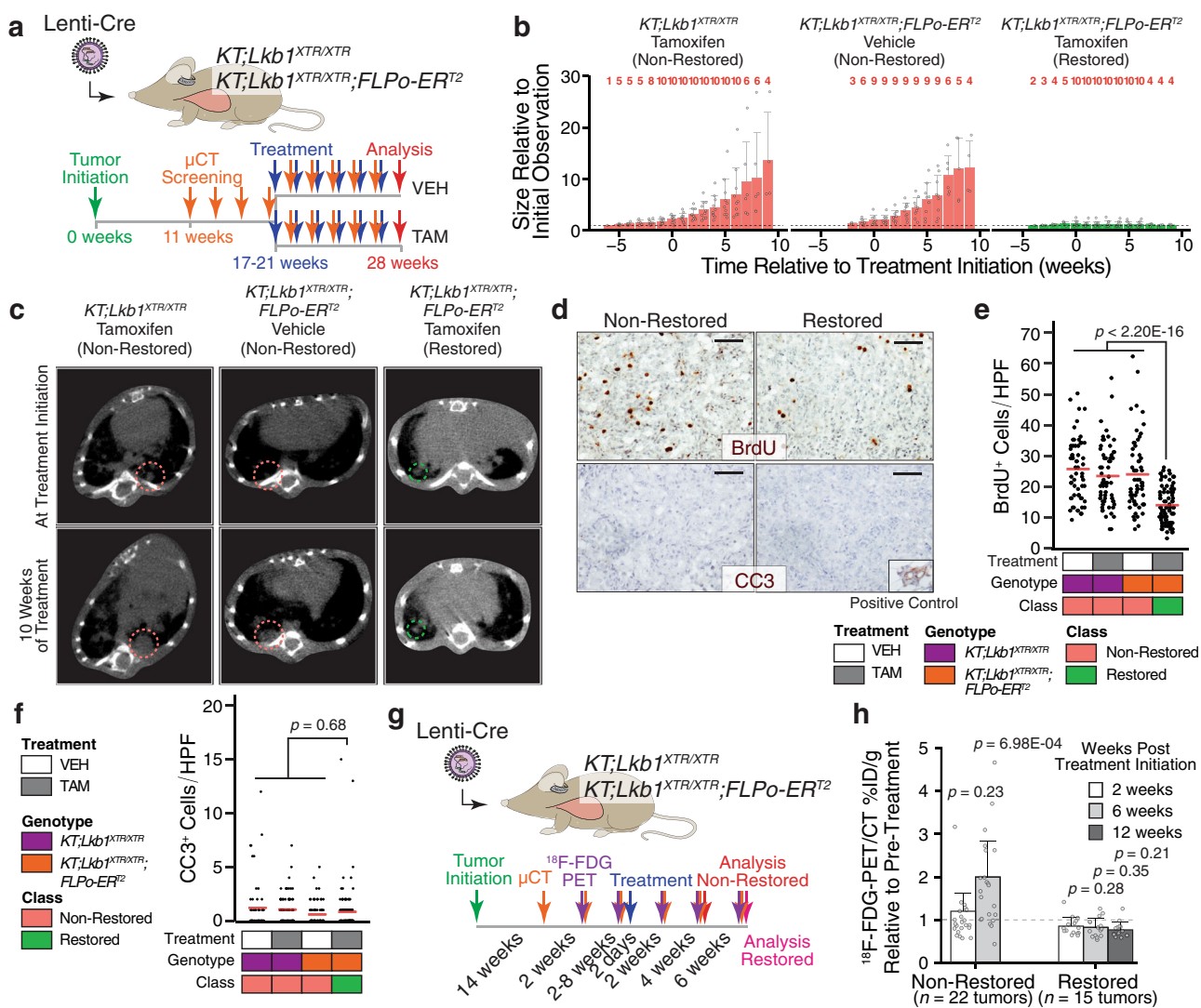

**Fig. 2 Lkb1 restoration drives tumor stasis and suppresses the increase in glucose avidity that accompanies progression. a** Longitudinal μCT imaging of lung tumors in *KT;Lkb1^XTR/XTR* and *KT;Lkb1^XTR/XTR;FLPo-ER^T2* mice. Treatment began within 1 to 6 weeks from initial detection of lung tumors. Tumors were tracked for an additional 6–10 weeks, and lung tissue was harvested at 28 weeks. VEH vehicle, TAM tamoxifen. **b** Changes in tumor volume. Red numbers indicate the number of tumors measured at a given time point. Bars correspond to the mean tumor volume relative to size at first detection. Error bars indicate standard deviation. Source data is displayed in Supplementary Fig. 6b. *KT;Lkb1^XTR/XTR*-tamoxifen, *n* = 10 tumors; *KT;Lkb1^XTR/XTR;FLPo-ER^T2*-vehicle, *n* = 9 tumors; *KT;Lkb1^XTR/XTR;FLPo-ER^T2*-tamoxifen, *n* = 10 tumors. **c** Representative μCT images of tumor-bearing lungs at treatment initiation (top) and after 10 weeks of treatment (bottom). **d** BrdU (top) and cleaved caspase 3 (CC3; bottom) detection by IHC within *Lkb1* non-restored (left) and restored (right) lung tumors following 2 weeks of treatment with vehicle or tamoxifen. Inset image shows CC3 staining of involuting mammary gland. Scale bars = 100 μm. Images were acquired from a single experiment including multiple biological replicates as noted in **e, f. e, f** Quantification of BrdU+ (**e**) and CC3+ (**f**) cells within *Lkb1*-restored and non-restored tumors. Each dot represents a ×20 field. Red crossbars indicate the mean. In **e**, *KT;Lkb1^XTR/XTR*-vehicle, *n* = 60 fields; *KT;Lkb1^XTR/XTR*-tamoxifen, *n* = 60 fields; *KT;Lkb1^XTR/XTR;FLPo-ER^T2*-vehicle, *n* = 60 fields; *KT;Lkb1^XTR/XTR;FLPo-ER^T2*-tamoxifen, *n* = 80 fields. In **f**, *KT;Lkb1^XTR/XTR*-vehicle, *n* = 60 fields; *KT;Lkb1^XTR/XTR*-tamoxifen, *n* = 80 fields; *KT;Lkb1^XTR/XTR;FLPo-ER^T2*-vehicle, *n* = 80 fields; *KT;Lkb1^XTR/XTR;FLPo-ER^T2*-tamoxifen, *n* = 120 fields. One tissue section per mouse was analyzed. *P* values calculated by two-sided unpaired *t* test. HPF high-power field, VEH vehicle, TAM tamoxifen. **g** Serial ^18F-FDG-PET/CT imaging. Tamoxifen treatment of *KT;Lkb1^XTR/XTR*, and *KT;Lkb1^XTR/XTR;FLPo-ER^T2* mice began within 2 days of establishing baseline levels of ^18F-FDG uptake (at least two measurements within 18–24 weeks after tumor initiation). ^18F-FDG uptake was captured after 2 and 6 weeks of treatment, as well as at 12 weeks for restored mice. **h** Changes in ^18F-FDG uptake in restored (*n* = 15 tumors) and non-restored (*n* = 22 tumors) tumors. Source data displayed in Supplementary Fig. 8c. *P* values calculated by two-sided unpaired *t* test.

and glycolysis-related genes, which agrees with our IHC and ^18F-FDG-PET/CT results (Supplementary Fig. 10b–d). Furthermore, genes relating to mTOR signaling were lower in restored tumors, which is consistent with a canonical function of LKB1 in inhibiting mTOR complex 1 via activation of AMPK (Supplementary Fig. 10e)[59]. Together, these initial findings highlight transcriptional changes reflecting reduced proliferation and altered metabolism in response to *Lkb1* restoration, as well as

increased expression of genes relating to specialized functions of ATII cells.

**Lkb1 restoration rescues features of alveolar type II identity.** To identify potential mediators of the transcriptional changes induced by *Lkb1* restoration, we performed motif enrichment analysis. Among the promoters of those genes that are higher

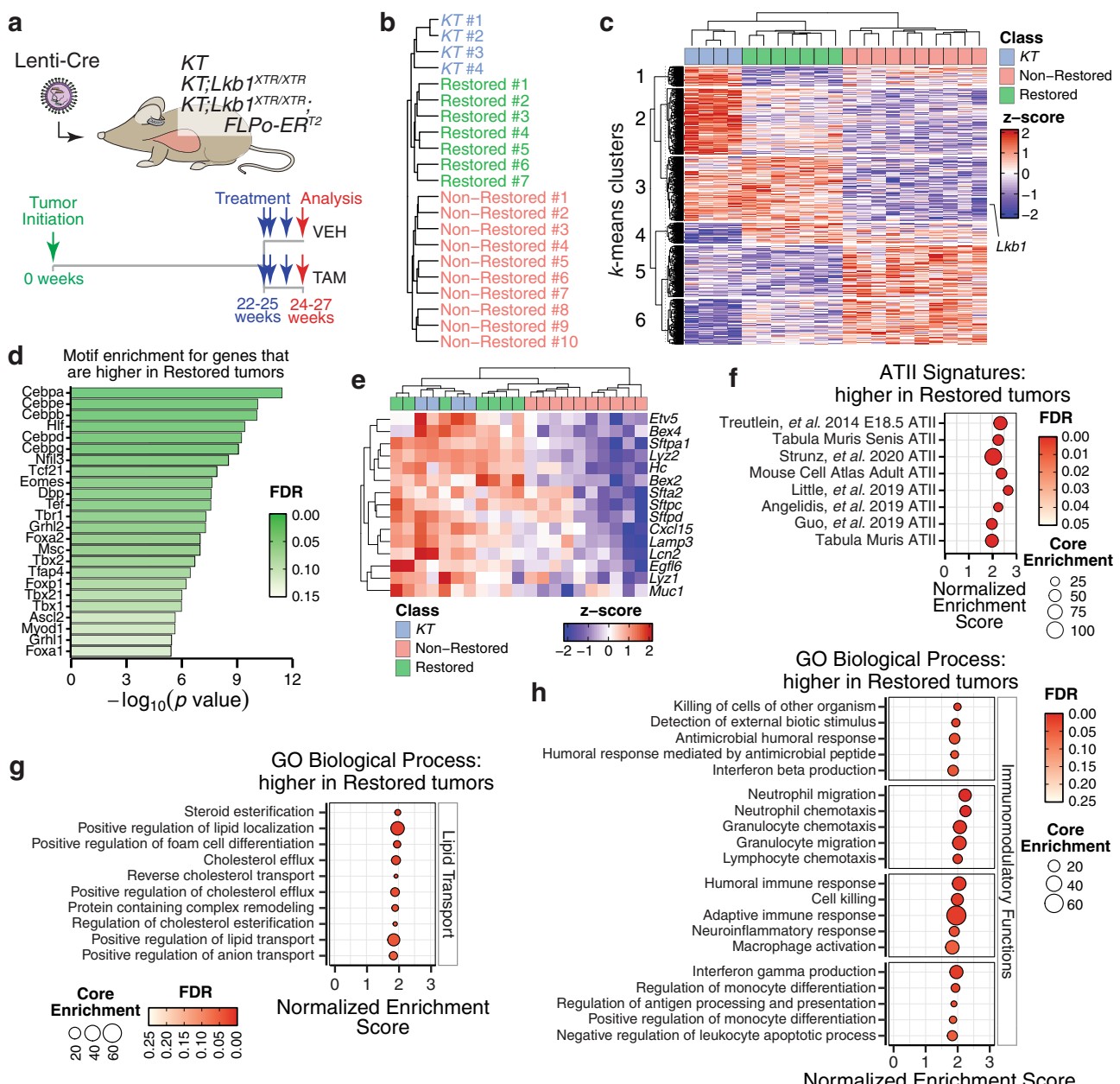

**Fig. 3 *Lkb1* restoration drives programs related to alveolar type II epithelial cell functions in lung adenocarcinoma. a** Profiling the acute transcriptional response to *Lkb1* restoration within established lung tumors. Mice were treated with vehicle or tamoxifen for 2 weeks prior to fluorescence-activated cell sorting (FACS)-isolation of tdTomato^positive neoplastic cells for RNA-seq analysis. KT, n = 4 mice; KT;Lkb1^{XTR/XTR}-tamoxifen, n = 3 mice; KT;Lkb1^{XTR/XTR};FLPo-ERT2-vehicle, n = 3 mice; KT;Lkb1^{XTR/XTR};FLPo-ERT2-tamoxifen, n = 4 mice. VEH vehicle, TAM tamoxifen. **b** Hierarchical clustering of the transcriptional profiles of *Lkb1* wild-type (KT), non-restored, and restored tumors by Euclidean distance. **c** Heatmap of genes that vary significantly (FDR < 0.05; likelihood ratio test). *Lkb1* is indicated. K-means clustering defined sets of genes that change concordantly across all samples (left). **d** Transcription factor motifs enriched within the putative promoters (−450 to +50 bp from transcription start site) of those genes that are higher (log₂ Fold Change >1 and FDR < 0.05) in restored lung tumors relative to non-restored using the JASPAR 2018 collection of position frequency matrices. **e** Expression of markers associated with alveolar type II epithelial cell identity across *Lkb1* wild-type, non-restored, and restored tumors. **f–h** GSEA using signatures of ATII identity derived from previously published single-cell RNA-seq datasets (**f**) and the Gene Ontology (GO) Biological Process module (**g, h**), illustrating the enrichment of gene sets among those genes that are higher in restored relative to non-restored tumors. Groups of enriched gene sets relating to lipid metabolism and transport (**g**) as well as immunomodulation (**h**) are shown. The size of the dots corresponds to the number of core enrichment genes and the fill color reflects FDR.

within restored tumors, there was a significant enrichment of C/EBP motifs (83 of the 128 LKB1-induced genes had C/EBPα motifs within their promoters) (Fig. 3d). Members of the C/EBP family of transcription factors coordinate proliferation and differentiation in multiple tissue contexts[60,61]. In particular, C/EBPα activity is required for ATII differentiation, suggesting that LKB1

may operate upstream of C/EBP factors to drive ATII differentiation[62,63]. Complementary to these findings, Sp/Klf motifs were enriched among genes that were higher in non-restored tumors (Supplementary Fig. 10f). Sp/Klf activity is enriched in alveolar epithelial progenitors and a regenerative subset of ATII cells, suggesting that *Lkb1* inactivation leads to loss

of ATII differentiation and reversion to a progenitor-like state[64,65].

Consistent with LKB1 maintaining ATII differentiation, restored tumors, like their *Lkb1* wild-type counterparts, had higher expression of several ATII markers relative to non-restored tumors, and multiple signatures of ATII identity were highly enriched in the restored state (Fig. 3e, f)[54,66–72]. Gene set enrichment analysis (GSEA) also revealed modest enrichment of signatures of ATII identity in *KT* tumors as compared to *Lkb1*-restored tumors and significant enrichment of signatures relating to morphogenesis within *Lkb1*-restored. This suggests that the non-overlapping nature of *Lkb1*-restored and *Lkb1* wild-type transcriptional states by PCA is attributable to varying degrees of ATII-like differentiation and the activity of broader developmental programs (Supplementary Fig. 10g, h). The induction of ATII markers by *Lkb1* restoration was conserved even in the absence of p53 (Supplementary Fig. 11a–f and Supplementary Data 1 and 5). Consistent with the role of SIKs as critical effectors of LKB1-mediated tumor suppression in the lung, mouse lung tumors with CRISPR/Cas9-mediated targeting of Siks also had lower expression of several ATII markers, suggesting that the LKB1-SIK axis maintains ATII identity (Supplementary Fig. 11g)[35,36]. Notably, a subset of ATII markers, including SFTPA1, CXCL15, LYZ2, and HC, were also higher at the protein level within restored tumors relative to non-restored tumors (Supplementary Fig. 12a–f and Supplementary Data 6 and 7). Taken together, these findings suggest that LKB1 maintains ATII identity and that *Lkb1* restoration induces features of ATII cells within established lung tumors.

Beyond specific markers of ATII identity, we also noted the upregulation of processes relating to ATII functions in response to *Lkb1* restoration. Gene sets pertaining to lipid metabolism and export as well as immunomodulation, were higher in restored tumors as compared to non-restored tumors at the mRNA level (Fig. 3g, h)[54]. At the protein level, lipid metabolism gene sets were enriched among the proteins that were more abundant in restored tumors, which is consistent with the lipid-processing functions required for surfactant production by mature ATII cells (Supplementary Fig. 12g). Additionally, there was an enrichment of mitochondrial proteins involved in oxidative phosphorylation among the proteins higher in restored tumors, which agrees with previous work demonstrating increased mitochondrial respiration capacity upon re-expressing *LKB1* in human lung cancer cells (Supplementary Fig. 12g)[59]. Furthermore, mitochondrial function is tightly linked to lipid metabolism in ATII cells, as mitochondria generate intermediates for the synthesis of phospholipids that are required for surfactant production[73,74]. Collectively, our transcriptomic and proteomic profiling of the acute response to *Lkb1* restoration in vivo indicates that *Lkb1* inactivation leads to the loss of ATII differentiation, which is rapidly reversible upon *Lkb1* restoration.

**The acute response to *Lkb1* restoration predominantly impacts the neoplastic epithelial compartment.** Extending from our observation that *Lkb1* restoration re-establishes features of ATII identity, we sought to understand whether *Lkb1* restoration modulates the cellular composition of lung tumors. To uncover changes in cellular state and/or abundance both within and outside of the neoplastic compartment, we performed single-cell RNA-seq on cells dissociated from lung tumors of *KT;Lkb1^{XTR/XTR}* and *KT;Lkb1^{XTR/XTR};FLPo-ER^{T2}* mice following 2 weeks of tamoxifen treatment (Supplementary Fig. 13a and "Methods"). Across all tumors, we observed diverse populations of immune, stromal, and neoplastic epithelial cells (Supplementary Fig. 13b, c). Apart from a fivefold change in the relative abundance of a

rare population of putative mast cells, short-term *Lkb1* restoration did not significantly alter the abundance of immune and stromal cell clusters (Supplementary Fig. 13d). The lack of change in the abundance of infiltrating neutrophil and T cells was notable given that *Lkb1* inactivation in lung tumors has been associated with increased neutrophil recruitment and decreased T cell infiltration[31]. Our results could stem from the relatively short time period after *Lkb1* restoration and/or be due to the fact that these indicators of an immunosuppressive microenvironment have been more clearly linked to the adenosquamous histotype, which does not develop in this mouse model[29,36,52].

To uncover gene expression changes within each cellular compartment that may reflect cell-state changes induced by *Lkb1* restoration, we collapsed each cluster into pseudobulk samples and performed differential gene expression analysis between restored and non-restored samples. Apart from a global increase in *Lkb1* due to the removal of the XTR gene trap within all cellular compartments, the most extensive gene expression changes occurred within the neoplastic epithelial compartment (Supplementary Fig. 13e–g). Focused examination of the neoplastic compartment revealed three sub-clusters, including ATI-like, ATII-like, and an "indeterminate" ATII-like subpopulation, which had attenuated expression of ATII markers (Supplementary Fig. 14a, b). Stratification of the neoplastic compartment on the basis of *Lkb1* status uncovered an enrichment of the indeterminate cluster within non-restored tumors and the ATII-like cluster in restored tumors (Supplementary Fig. 14c). Thus, in agreement with our bulk analyses, *Lkb1* activity appears to drive neoplastic cells into a mature ATII cell state.

**Neoplastic cells exist across a cell state spectrum resembling the progression from ATII to ATI identities.** To more thoroughly interrogate the cell states within the neoplastic compartment, we performed single-cell RNA-seq on tdTomato^{positive} neoplastic cells sorted from restored and non-restored lung tumors (Fig. 4a and Supplementary Fig. 15a). Across all tumors, we observed four major clusters of cells and a minor cluster of highly proliferative cells (Fig. 4b, c). The largest cluster expressed markers of ATII cells, such as *Lyz2*, *Sftpa1*, *Hc*, and *Cxcl15* (Fig. 4c)[54,75]. As noted in our initial single-cell analysis, there also existed an "indeterminate" ATII-like cluster. The remaining two clusters resembled the ATI state, with the larger of the two expressing established ATI markers, including *Ager* and *Hopx* (Fig. 4c)[54,75]. In contrast, the smaller ATI-like cluster had high expression of *Krt8* and *Krt19*, which delineate a "stalled" ATII-ATI transitional state that emerges following lung injury (Fig. 4c)[64,66,76].

To elucidate the relationship between the indeterminate subpopulation and the other subpopulations, we performed dynamic inference analyses. Both RNA velocity analysis and pseudotemporal ordering suggested that the indeterminate cluster arises from the ATII-like cluster and represents an intermediate state along with the progression from ATII to ATI states, which resembles ATII-ATI trans-differentiation that occurs in response to lung injury (Fig. 4d, e)[64,66,76]. Consistent with the Krt8^+/Krt19^+ population representing a stalled transitional state during ATII-ATI trans-differentiation, this cluster branches off from the ATII-ATI primary trajectory (Fig. 4e and Supplementary Fig. 15b)[66,76]. These findings indicate that the neoplastic compartment comprises several identities that resemble the spectrum of states that emerge during ATII-ATI trans-differentiation, including an indeterminate subpopulation that likely arises from the ATII-like population and may represent an intermediate transition state.

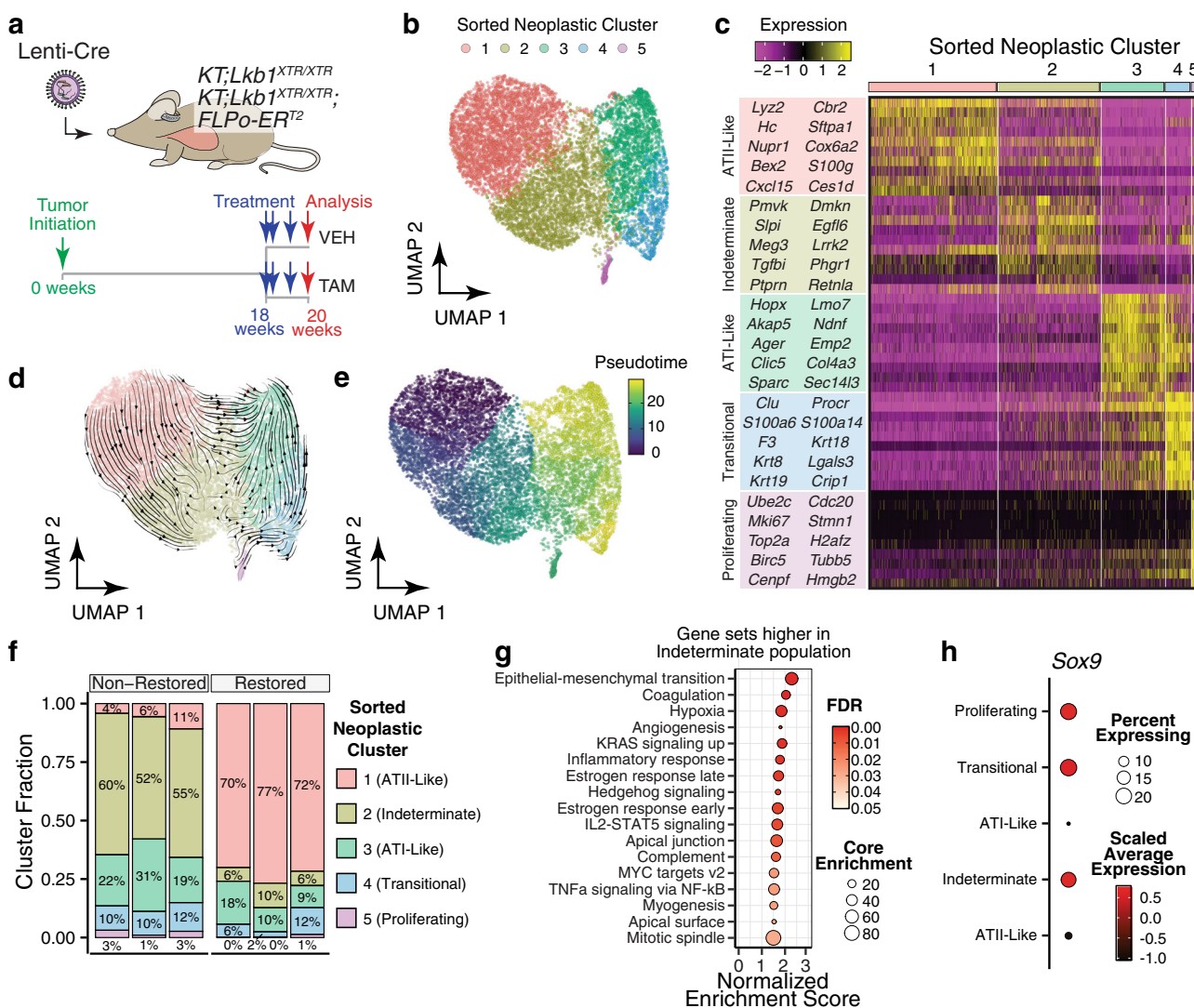

**Fig. 4 Lkb1 restoration enforces an alveolar type II-like cell state. a** Profiling the acute transcriptional response to *Lkb1* restoration within established tumors at single-cell resolution. *KT;Lkb1^XTR/XTR*-vehicle, n = 1 mouse; *KT;Lkb1^XTR/XTR*-tamoxifen, n = 1 mouse; *KT;Lkb1^XTR/XTR*;*FLPo-ER^T2*-vehicle, n = 1 mouse; *KT;Lkb1^XTR/XTR*;*FLPo-ER^T2*-tamoxifen, n = 3 mice. **b** Single-cell RNA-seq on tdTomato^positive neoplastic cells from *Lkb1*-restored and non-restored tumors. Cell fill reflects assignment to clusters defined by the Louvain algorithm. **c** Top ten markers that define each of the neoplastic cell clusters. The predicted cell type identities are listed to the left. Louvain cluster assignments are indicated by the bars at the top of the heatmap. **d** RNA velocity analysis on neoplastic cells. Velocity vectors are summarized as streamlines overlaid on UMAP embeddings. The general direction of flow follows that of the ATII-ATI differentiation axis, with the indeterminate cluster residing at an intermediate position. **e** Trajectory inference analysis on sorted neoplastic cells. Cell fill corresponds to its relative position in pseudotime. The pseudotime trajectory runs along the ATII-ATI differentiation axis, with the indeterminate cluster residing at an intermediate position. **f** Proportion of each neoplastic epithelial subpopulation within *Lkb1*-restored and non-restored tumors. Fill reflects Louvain cluster assignments, and their corresponding predicted identities are listed. Each column corresponds to an individual animal. **g** GSEA comparing cells of the ATII-like and indeterminate clusters at the pseudobulk level using the Hallmarks module. The size of dots corresponds to the number of core enrichment genes and the fill color reflects FDR. The plotted gene sets correspond to those that were enriched among the genes that are higher in the indeterminate cluster relative to the ATII-like cluster. **h** Expression of *Sox9* across the five Louvain clusters. Predicted cluster identities are listed (left). Fill indicates average expression and dot size reflects the proportion of cells within a given cluster that express *Sox9*.

**LKB1 drives neoplastic cells toward an ATII-like cell state.** Upon stratification of the scRNA-seq dataset into *Lkb1*-restored or non-restored tumors, we uncovered a striking shift from the indeterminate state within non-restored tumors to the ATII-like state within restored tumors (a 14-fold increase in the proportion of ATII-like cells and an 8-fold reduction in indeterminate cells within restored tumors as compared to non-restored tumors) (Fig. 4f). We also observed significant concordance between our single-cell and bulk RNA-seq datasets in terms of the gene expression changes induced by *Lkb1* restoration (Supplementary Fig. 15c, d). Notably, relative to restored tumors, the proportion

of actively proliferating cells identified by elevated expression of genes relating to cell cycle progression was greater within non-restored tumors, and cells derived from non-restored tumors were significantly over-represented within the actively pro-liferating population (Fig. 4f and Supplementary Fig. 15e). Upon regression of variation due to cell cycle-driven gene expression changes, we observed that the ATII-like subpopulation was under-represented among cells that were initially identified as actively proliferating, suggesting that the ATII-like state is less proliferative relative to the other subpopulations (Supplementary Fig. 15f). Notably, the fraction of proliferative cells was reduced in

*Lkb1*-restored tumors as compared to non-restored tumors across each of the subpopulations, suggesting that the growth-suppressive response to LKB1 activity extends beyond the indeterminate state (Supplementary Fig. 15g). Collectively, these analyses indicate that *Lkb1* restoration drives the transition from an indeterminate state to a more differentiated and less proliferative ATII-like state.

To identify molecular features that distinguish the indeterminate state and potentially uncover the mechanism by which it emerges as a consequence of *Lkb1* inactivation, we compared the gene expression profiles of the ATII-like and indeterminate clusters (Supplementary Fig. 15h). GSEA revealed that gene sets relating to proliferation, EMT, hypoxia, and KRAS signaling were higher in the indeterminate subpopulation (Fig. 4g). In agreement with our bulk RNA-seq analysis, C/EBP motifs were enriched within the promoters of genes that are higher in the ATII-like cluster, as well as an enrichment of ATII signatures, indicating that the indeterminate cluster represents a C/EBP-low state lacking features of ATII differentiation (Supplementary Fig. 15i, j)[54,66–72]. Furthermore, Sp/Klf motifs and signatures of ATI identity were enriched among the genes that were higher in the indeterminate cluster, consistent with the indeterminate population representing a progenitor-like state that arises as a consequence of the loss of ATII differentiation (Supplementary Fig. 15j, k)[64,65]. *Sox9*, which is a marker delineating distal tip progenitor cells that give rise to the alveolar lineages and also an inhibitor of alveolar differentiation, was more highly expressed in the indeterminate cluster as compared to the ATII-like state (Fig. 4h)[77,78]. The indeterminate subpopulation also had increased *Wnt5a* expression, which has been shown to potentiate the mitogenic activity of epidermal growth factor in ATII cells (Supplementary Fig. 15l)[79]. Taken together, these findings demonstrate that *Lkb1* restoration reinstates ATII-like differentiation, driving cells away from a C/EBP-low, progenitor-like state that emerges as a consequence of *Lkb1* inactivation.

**C/EBP transcription factors suppress lung tumor growth**. To investigate whether C/EBP transcription factors and a subset of genes exhibiting LKB1-dependent expression function as tumors suppressors, we integrated CRISPR/Cas9 with tumor barcoding and high-throughput barcode sequencing (Tuba-seq) to assess the impact of inactivating candidate genes on tumor growth[42]. Of the six members within the C/EBP family, we focused on those paralogs that are most highly expressed within oncogenic KRAS-driven lung tumors (Supplementary Fig. 16a), excluding the inhibitory paralog, C/EBPγ (*Cebpg*), and C/EBPζ (*Ddit3*), which redirects C/EBP activity from canonical target genes[80–83]. To inactivate each candidate gene in a multiplexed format, we initiated tumors in *KT* and *KT;H11^LSL-Cas9^* mice using a pool of Lenti-sgRNA/Cre vectors that include two-component barcodes comprised of sgRNA and clonal identifiers (sgID-BC) (Fig. 5a and Supplementary Fig. 16b)[42]. Fourteen weeks after tumor initiation, we quantified distributions of tumor size across each genotype by deep sequencing of the sgID-BC region that had been amplified from the integrated lentiviral genomes within bulk tumor-bearing lungs (Supplementary Fig. 16b)[42]. Strikingly, simultaneous targeting of *Cebpa*, *Cebpb*, and *Cebpd* significantly increased tumor growth, while the inactivation of other LKB1-dependent genes had no significant impact on tumor growth (Fig. 5b and Supplementary Fig. 16c). In conjunction with our observations that C/EBP activity is increased upon *Lkb1* restoration, these findings suggest that C/EBP transcription factors may be critical effectors of LKB1-mediated tumor suppression.

To validate the tumor-suppressive capacity of C/EBP factors, we initiated tumors in *KT* and *KT;H11^LSL-Cas9^* mice with either Lenti-sgNeo1-sgNT-sgNeo2/Cre (Lenti-sgInert/Cre) or Lenti-

sg*Cebpa*-sg*Cebpb*-sg*Cepbd*/Cre (Lenti-sg*Cebpa/b/d*/Cre) (Fig. 5c). CRISPR/Cas9-mediated targeting of C/ebp factors increased overall tumor burden in terms of the total tumor area and individual tumor size (Fig. 5d–f and Supplementary Fig. 16d). Together, these data indicate that C/EBP factors constrain oncogenic KRAS-driven lung tumor growth in vivo.

**Inactivation of C/ebp transcription factors recapitulates transcriptional features of *Lkb1* deficiency**. Given that genes induced by LKB1 activity were enriched with C/EBP motifs, we determined the extent to which the transcriptional changes elicited by *Lkb1* inactivation can be attributed to reduced C/EBP activity. To compare the transcriptional profiles of *Lkb1*- and C/ebp-targeted tumors, we performed RNA-seq on neoplastic cells sorted from tumors initiated in *KT;H11^LSL-Cas9^* mice with Lenti-sgInert/Cre, Lenti-sg*Cebpa/b/d*/Cre, or Lenti-sg*Lkb1*/Cre (Supplementary Fig. 17a and Supplementary Data 8–11). Principal component analysis and hierarchical clustering separated C/ebp- and *Lkb1*-targeted tumors from tumors initiated with sgInert, suggesting conserved transcriptional changes in C/ebp- and *Lkb1*-targeted tumors (Supplementary Fig. 17b, c). Among the genes that vary significantly across the dataset, we defined six groups by *k*-means clustering (Fig. 6a). The genes that were higher in both C/ebp- and *Lkb1*-targeted tumors relative to sgInert tumors (354 of the 1823 variable genes) were enriched for genes relating to extracellular matrix interactions, migration, adhesion, and respiratory development (Fig. 6a and Supplementary Fig. 17d). Conversely, the genes that were lower in C/ebp- and *Lkb1*-targeted tumors (352 of the 1823 variable genes) were enriched for genes relating to translation and sterol biosynthesis (Fig. 6a and Supplementary Fig. 17d). Notably, several markers of ATII identity were lower in C/ebp-targeted and *Lkb1*-targeted tumors, consistent with the loss of ATII differentiation upon inactivation of C/ebp factors or *Lkb1* (Fig. 6a). Furthermore, by GSEA, we found that C/EBP-dependent genes were enriched among those genes that were higher in the ATII-like neoplastic subpopulation (Fig. 6b). Moreover, genes that were higher in C/ebp-targeted tumors were enriched among the genes that were higher within the indeterminate population, thus reinforcing the notion that the indeterminate state corresponds to progenitor-like cells exhibiting low C/EBP activity (Fig. 6b). These analyses uncovered shared transcriptional changes upon inactivation of either C/ebp factors or *Lkb1*, suggesting that C/EBPs may operate downstream of LKB1 to suppress tumor growth and maintain ATII identity.

To further examine the extent of conservation of transcriptional changes upon inactivation of either C/ebp factors or *Lkb1*, we directly compared differentially expressed genes and pathways relative to sgInert tumors. In both C/ebp- and *Lkb1*-targeted contexts, there was a highly significant overlap in terms of genes that were lower or higher relative to tumors initiated with sgInert (Fig. 6c, d and Supplementary Fig. 17e). Among the genes that were lower in either C/ebp- or *Lkb1*-targeted tumors, there was a significant enrichment of genes with upstream C/EBP motifs (with 76 out of 121 promoters of LKB1-dependent genes and 121 out of 230 promoters of C/EBP-dependent genes containing C/EBPα motif), consistent with our previous observation of C/EBP activity downstream of LKB1 (Supplementary Fig. 17f). C/ebp- and *Lkb1*-targeted tumors also exhibit overlap in terms of enriched gene sets, particularly those relating to adhesion, migration, and extracellular matrix interaction (Supplementary Fig. 17g). In conjunction with increased expression of putative C/EBP target genes in response to *Lkb1* restoration, these findings indicate that C/ebp inactivation elicits transcriptional changes that resemble *Lkb1* deficiency and suggest that C/EBPs function downstream of LKB1.

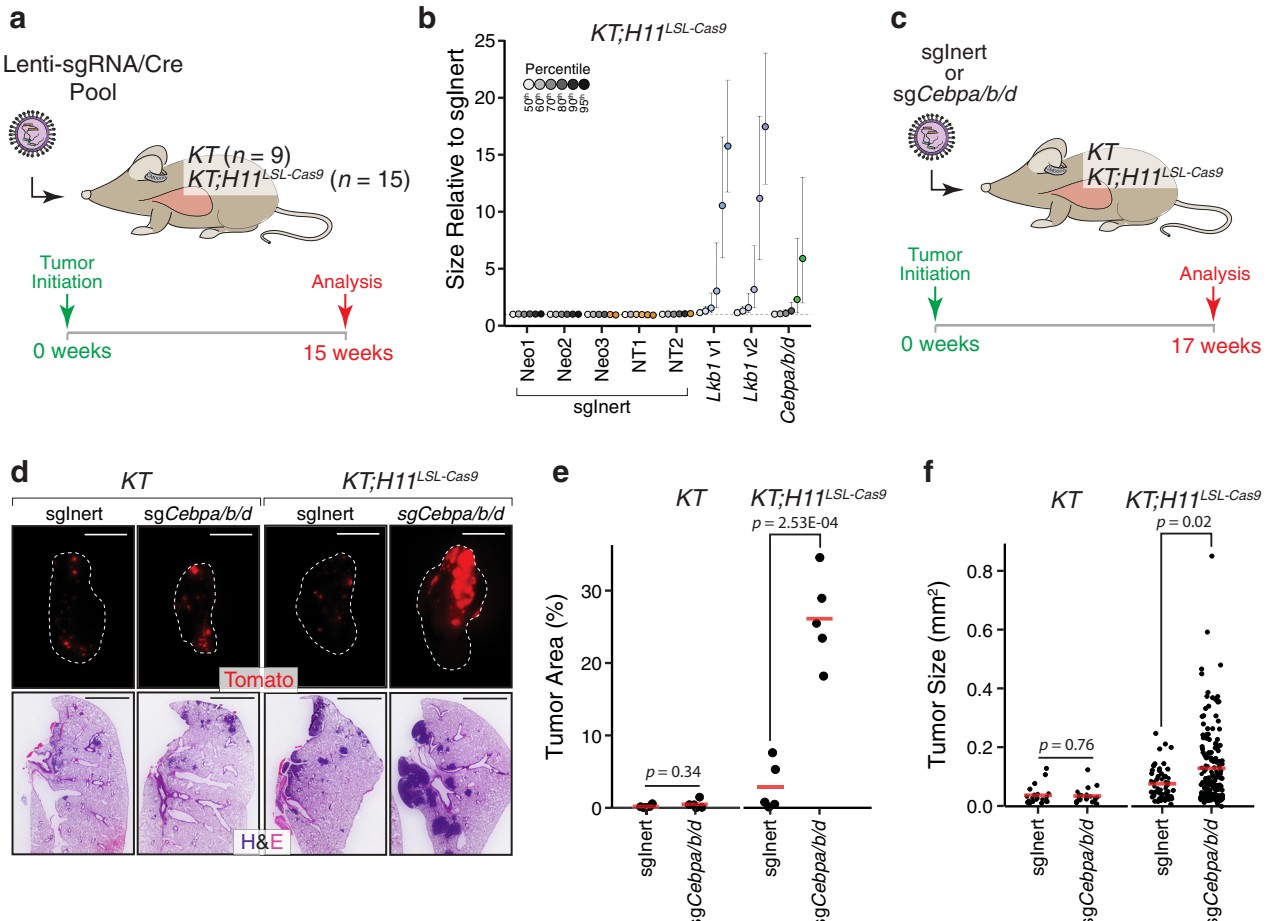

**Fig. 5 C/EBP transcription factors constrain oncogenic KRAS-driven lung tumor growth. a** Interrogation of the tumor-suppressive capacity of a series of LKB1-dependent genes and C/EBP transcription factors (targets listed in Supplementary Fig. 16b). Lenti-sgRNA/Cre vectors targeting each candidate gene were pooled prior to delivery into *KT* and *KT;H11^LSL-Cas9* mice (method outlined in Supplementary Fig. 16b). **b** Bulk tumor-bearing lungs were analyzed by Tuba-seq. Percentile plot depicting tumor size at several percentiles relative to the distribution of tumors initiated with Lenti-sgRNA/Cre vectors encoding inert sgRNAs (sgNeo1, sgNeo2, sgNeo3, sgNT1, sgNT2). *Lkb1* and *C/ebp* family targeting vectors are shown. Each vector has a distinct fill color, and fill saturation indicates percentile. Colored fill indicates that tumor size at a given percentile is significantly different from inert sgRNAs, while grayscale indicates no significant difference. Error bars indicate 95% confidence intervals centered on the mean of relative tumor size at a given percentile. **c** Validation of C/EBP factors as suppressors of oncogenic KRAS-driven tumor growth in a non-multiplexed format. Tumors were initiated in *KT* and *KT;H11^LSL-Cas9* mice with either Lenti-sgInert/Cre (sgNeo1/sgNT1/sgNeo2; sgInert) or Lenti-sg*Cebpa/b/d*/Cre (sg*Cebpa*/sg*Cebpb*/sg*Cebpd*; sgCebpa/b/d). *N* = 5 mice per genotype-virus cohort. **d** Representative fluorescence (top) and H&E (bottom) images of tumor-bearing lungs from *KT* and *KT;H11^LSL-Cas9* mice transduced with either Lenti-sgInert/Cre or Lenti-sg*Cebpa/b/d*/Cre. Lung lobes within fluorescent images are outlined in white. *N* = 5 mice per genotype-virus cohort. Top scale bars = 5 mm. Bottom scale bars = 2 mm. **e**, **f** Quantification of tumor area (**e**) and tumor size (**f**) by histological examination of tumor-bearing lungs from *KT* and *KT;H11^LSL-Cas9* mice transduced with either Lenti-sgInert/Cre or Lenti-sg*Cebpa/b/d*/Cre. Each dot represents either individual mice (**e**) or individual tumors (**f**). Red crossbars indicate the mean. In **e**, *n* = 5 mice per genotype-virus cohort. One tissue section per mouse was analyzed. In **f**, *KT*-sgInert, *n* = 22 tumors; *KT*-sg*Cebpa/b/d*, *n* = 23 tumors; *KT;H11^LSL-Cas9*-sgInert, *n* = 58 tumors; *KT;H11^LSL-Cas9*-sg*Cebpa/b/d*, *n* = 167 tumors. *P* values were calculated by a two-sided unpaired *t* test.

**A subset of LKB1- and C/EBP-dependent genes are NKX2-1 target genes.** While there was significant concordance in the gene expression changes elicited by inactivating either *Lkb1* or C/ebp factors, not all genes were regulated by both C/EBPs and LKB1, suggesting more nuanced regulation of C/EBP targets by LKB1 signaling. Comparison with the gene expression changes elicited by Sik targeting indicated significant but incomplete overlap with those changes induced by C/ebp targeting, suggesting that the link between C/EBPs and the LKB1-SIK axis is unlikely to be strictly linear (Supplementary Fig. 17h). To identify potential transcription factors downstream of LKB1 that could cooperate with C/EBPs, we performed motif enrichment on subsets of genes that were uniquely or jointly dependent on LKB1 or C/EBPs. Interestingly, among those genes that were jointly dependent on LKB1 and C/EBPs, there was a significant enrichment for motifs belonging to TATA-binding

protein, bHLH factors (FIGLA and MSC), the HNF family, as well as the NKX2 family (33 out of 51 promoters analyzed had NKX2 motifs) (Fig. 6e). C/EBPα has been proposed to direct ATII-specific, NKX2-1-driven transcriptional programs within the distal lung epithelium[84]. Consistent with the concerted action of C/EBPα and NKX2-1, C/EBP motifs were also highly enriched at NKX2-1-bound sites from previous ChIP-seq data from oncogenic KRAS-driven lung tumors (Supplementary Fig. 17i)[85]. Furthermore, NKX2-1 binds proximal to 86% of genes that are dependent on both LKB1 and C/EBP (Fig. 6f). Of the C/EBP- and LKB1-dependent, NXK2-1-bound genes, nearly half exhibit NKX2-1-dependent expression, suggesting that NKX2-1 may cooperate with C/EBP factors to drive a subset of LKB1-dependent genes (Fig. 6g)[86]. Notably, among this subset were several markers of ATII identity, including *Etv5*, *Cxcl15*, *Hc*, *Lyz2*, thus underscoring the role of the

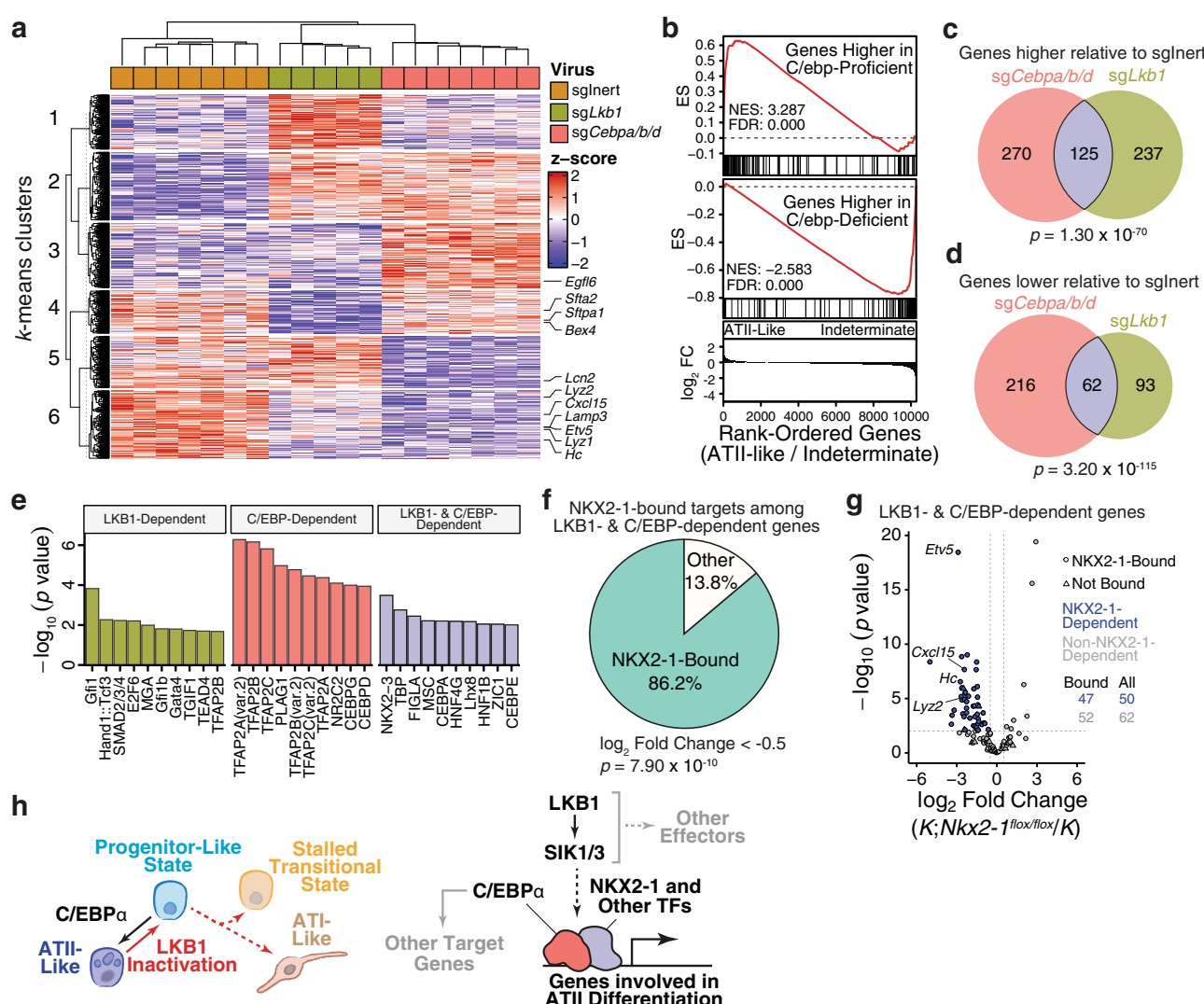

**Fig. 6 C/EBP transcription factors co-regulate a subset of LKB1-dependent genes. a** Transcriptional profiling of *Lkb1*- and C/ebp-targeted tumors as well as oncogenic KRAS-only tumors (sgInert). Genes that vary significantly (FDR < 0.05; likelihood ratio test). Markers of alveolar type II identity are indicated. **b** Enrichment of C/ebp-dependent genes (top) among genes that are higher in the ATII-like state. Enrichment of genes that are higher in C/ebp-targeted tumors (bottom) among genes that are higher in the indeterminate state. ES Running enrichment score. NES normalized enrichment score, FC fold change. **c, d** Comparison of genes that are higher (**c**) or lower (**d**; absolute log₂ Fold Change >1 and FDR < 0.05) in *Lkb1*- or C/ebp-targeted tumors relative to tumors driven by oncogenic KRAS alone (sgInert). *P* values from hypergeometric tests are indicated. **e** Transcription factor motif enrichment on genes that are either uniquely lower (log₂ Fold Change < −1 and FDR < 0.05) in *Lkb1*- (left) or C/ebp-targeted tumors (middle) relative to sgInert tumors or are jointly LKB1- and C/EBP-dependent (right). **f** Proportion of genes proximal to NKX2-1 binding sites (within −3 to +1 kb from TSS) among those that are jointly dependent upon LKB1 and C/EBPs (log₂ Fold Change < −0.5 and FDR < 0.05). *P* value from hypergeometric test is indicated. Derived from previously published NKX2-1 ChIP-seq dataset[85]. **g** Transcriptional comparison of *Nkx2-1*-deficient and *Nkx2-1*-proficient lung tumors using previously published gene expression data[86]. Genes that are jointly dependent upon LKB1 and C/EBPs (log₂ Fold Change < −0.5 and FDR < 0.05) are plotted. Blue fill denotes genes that are dependent upon NKX2-1(log₂ Fold Change < −0.5 and FDR < 0.05). Shape indicates proximal binding of NKX2-1 (within −3 to +1 kb from TSS). The number of NKX2-1-bound genes that are or are not NKX2-1-dependent is indicated. **h** Proposed model for the function of LKB1 in lung tumors. *Lkb1* inactivation enables the emergence of neoplastic cell states outside of ATII-like identity. Upon *Lkb1* restoration, progenitor-like cells assume a more mature, less proliferative ATII-like identity, reflecting the increased activity of the lineage-defining factor, C/EBPα (left). At the molecular level (right), C/EBP operates indirectly downstream of LKB1-SIK in cooperation with NKX2-1 and other transcription factors to drive ATII differentiation.

concerted action of LKB1 and C/EBPs in maintaining ATII identity. Deconvolution of C/EBP-mediated tumor suppression through a secondary Tuba-seq screen indicated no evidence of functional redundancy among the C/EBP paralogs, with C/EBPα being the dominant tumor-suppressive factor (Supplementary Fig. 18a–d). Together, these findings indicate that NKX2-1 and likely other transcription factors mediate a functional link between LKB1 and C/EBPα in suppressing lung tumor growth and maintaining ATII differentiation.

## Discussion

Although tumor suppressor loss represents a major class of genetic alterations in cancer, the direct characterization of the function of these inactivated genes in vivo remains challenging. The advent of strategies to reversibly inactivate tumor suppressor function in vivo within mouse models has enabled the establishment of causal relationships between tumor suppressors and the physiological programs that they govern[15]. Here, we employed the XTR system to identify processes regulated by

LKB1, which is among the most frequently altered tumor suppressors in human lung cancer and one of the most potent suppressors of oncogenic KRAS-driven lung tumor growth[19,42,87]. To complement recent targeted efforts that identified SIKs as potent tumor-suppressive effectors among the direct substrates of LKB1 kinase activity, we leveraged reversible inactivation to identify LKB1-driven processes in an unbiased fashion[35,36].

Previous studies on tumor suppressor function in vivo have illuminated diverse responses to tumor suppressor restoration in vivo. *Trp53* restoration drives regression of malignant lung adenocarcinomas but has little effect in early adenomas[88]. In contrast, *Rb1* restoration blocks lung tumor progression to more advanced grades, impairs metastatic progression, and transiently inhibits tumor growth[18]. Strikingly, *Apc* restoration in colorectal tumors drives complete regression and restoration of normal tissue function, even in the context of oncogenic KRAS and concomitant *Trp53* inactivation[20]. Despite these examples, it remains plausible that the dependency on the absence of a tumor suppressor could diminish during tumor progression, thereby yielding established tumors that are insensitive to tumor suppressor reconstitution[6]. In fact, recent work has demonstrated that *LKB1* rescue in a subset of human lung cancer cell lines is insufficient to revert stable epigenetic changes that stem from *LKB1* loss[29]. However, in our experiments, *Lkb1* restoration suppressed proliferation and stably blocked the growth of established early-stage tumors, even over relatively long periods following *Lkb1* restoration (Figs. 1c–e and 2b–e and Supplementary Figs. 3a, b and 6b–d). Furthermore, this growth-suppressive effect was conserved in the allograft setting as well as in lung cancer cell lines in vitro (Supplementary Figs. 4 and 7).

There may be more nuance to the response to *Lkb1* restoration in vivo, including tumor stage or genotype specificity. Varying the timing of *Lkb1* restoration throughout tumor development (including within metastases) and the generation of concomitant genetic alterations will be critical to extend the clinical applicability of future studies. For instance, p53 loss dampens the effect of *Lkb1* restoration on early tumor growth, which could stem from accelerated tumor progression prior to *Lkb1* restoration or implicate p53 as an effector involved in LKB1-driven growth suppression (Supplementary Fig. 5). Beyond permuting temporal and genetic variables, a greater understanding of the mechanisms by which LKB1 inactivation imparts epigenetic plasticity may aid in the distinction of subsets of LKB1-deficient lung cancers or neoplastic cell states therein that remain sensitive to LKB1 activity[29].

*Lkb1* inactivation in oncogenic KRAS-driven lung tumors can lead to the emergence of diverse histological subtypes[25,36]. This expanded histological spectrum could stem from *Lkb1* inactivation either enabling tumor outgrowth from distinct cells of origin or imparting plasticity to allow for the acquisition of alternative differentiation states[59]. Here, we demonstrate that the disruption of *Lkb1* reduces C/EBP activity and enables the departure of neoplastic cells from ATII fate, thus potentially enhancing the propensity to assume alternative differentiation states (Fig. 6h). In addition to enabling the development of mucinous lung adenocarcinoma, *LKB1* loss is associated with increased expression of markers of gastric differentiation, such as *TFF1* and *MUC5AC* in human and mouse lung cancer[34,36,43,52,89,90]. Notably, the transition from ATII to gastric differentiation also results from the loss of NKX2-1 activity[84,85]. Like C/EBPα, NKX2-1 is critical for lung development and regulates ATII-associated processes like surfactant production, suggesting that LKB1 and C/EBPα may operate in concert with NKX2-1 to enforce ATII differentiation[62,84,91–96]. We demonstrate that NKX2-1 binds

proximal to and is required for the expression of about half of the genes that are jointly dependent on LKB1 and C/EBPs. This is consistent with a model in which C/EBPs cooperate with NKX2-1 and other transcription factors downstream of LKB1 to promote the activation of genes involved in ATII differentiation (Fig. 6h). Thus, disruption of this program may facilitate the departure from a differentiated state and reversion to a progenitor-like state.

The precise mechanism(s) by which LKB1 is linked to C/EBP transcriptional activity remains to be uncovered. C/EBPα appears to be the dominant tumor-suppressive paralog, with no evidence of functional redundancy (Supplementary Fig. 18c, d). This is consistent with the non-overlapping functions of C/EBP factors during development in addition to previous reports of a tumor-suppressive role for C/EBPα within the lung and a more general role in the inhibition of cell cycle progression[63,97–103]. Notably, *CEBPA* expression is frequently down-regulated in human non-small cell lung cancer due to promoter methylation or genetic deletion[99,100,102]. Within *Lkb1*-restored tumors, we noted a modest, yet significant, increase in *Cebpa* mRNA levels as compared to non-restored tumors, as well as a modest decrease in expression of the inhibitory paralog *Cebpg*, either of which could contribute to enhanced expression of C/EBPα targets (Supplementary Fig. 16a)[82]. SIK1/3 are key effectors of LKB1-mediated tumor suppression in lung cancer, and Sik-targeted tumors also exhibit lower expression of ATII markers relative to tumors driven by oncogenic KRAS alone, suggesting that SIKs may mediate the activation of the C/EBP-driven ATII differentiation program downstream of LKB1 (Fig. 6h and Supplementary Fig. 11g)[35,36]. Canonical targets of SIK activity are CRTC transcriptional coactivators and class IIa HDACs, both of which are inhibited by nuclear exclusion driven by SIK-mediated phosphorylation[59]. Notably, HDAC3 physically associates with NKX2-1, and both C/EBP and NKX2-1 motifs are enriched at HDAC3-bound sites in *Lkb1*-deficient lung cancer cells[104]. Thus, SIK-mediated regulation of HDAC trafficking could modulate the activity of C/EBPα and NKX2-1 targets via interference with corepressor complex recruitment[104–106].

Beyond the potential for a direct link between LKB1 signaling and activation of C/EBP target genes, the elevated expression of C/EBP targets and features of ATII differentiation within *Lkb1*-restored tumors could also be a product of indirect, selective or adaptive mechanisms. For instance, *Lkb1* restoration could drive selection for lowly cycling cells that retain ATII features by promoting cell death among proliferative, less-differentiated cells. However, our analyses of markers of proliferation and cell death in both cell lines in vitro and tumors in vivo most strongly support a model in which *Lkb1* restoration suppresses proliferation and drives reprogramming rather than promoting shifts in cell-state representation via selective induction of cell death (Fig. 2d–f and Supplementary Fig. 7). Alternatively, the induction of C/EBP targets and features of ATII differentiation could be an adaptive response independent of the LKB1-SIK axis of tumor suppression that facilitates the persistence of neoplastic cells when challenged by the reactivation of LKB1 signaling. Notably, AMPK has been shown to be required for endodermal fate specification during embryoid body formation, thus it is plausible that the enforcement of ATII differentiation could be attributed in part to the restoration of AMPK activity[107]. Future work centering on the elucidation of the pathways that govern C/EBP activity in lung epithelial cells will be critical to uncover a direct connection between LKB1 activity and the orchestration of ATII differentiation.

Through the implementation of our *Lkb1*[XTR] allele, we have uncovered a role for LKB1 at the interface of differentiation enforcement and proliferative control in lung cancer in addition to demonstrating that established lung tumors remain sensitive to

its tumor-suppressive activity. Beyond the context of lung tumors, we envision that the $Lkb1^{XTR}$ allele will enable the discovery of additional functions of LKB1 in other malignant and even normal contexts that would not have otherwise been identified through cell culture or in vivo systems involving constitutive inactivation of $Lkb1$. The ability to control LKB1 function in vivo should be particularly useful in assessing its role in broader physiological processes such as microenvironmental interactions and metabolic control. Overall, a deeper understanding of the direct functions of LKB1 in vivo will better inform how best to approach pharmacologically countering the molecular consequences of $LKB1$ loss in human cancer.

## Methods

**Ethics statement**. Mice were maintained within Stanford University's SIM1 Barrier Facility according to practices prescribed by the NIH and the Institutional Animal Care and Use Committee at Stanford University. Additional accreditation of Stanford University Research Animal Facility was provided by the Association for Assessment and Accreditation of Laboratory Animal Care. Protocols employed in this study were approved by the Administrative Panel on Laboratory Animal Care at Stanford University (Protocol #26696).

**Generation of $Lkb1^{XTR}$ targeting vector**. Right and left homology arms were amplified from the first intron of the $Lkb1$ locus and inserted at PacI-AatII and FseI-AscI sites of pNeoXTR f2 (Addgene #69159), respectively. Amplification of the $Lkb1$ right homology arm and addition of flanking 5'PacI and 3'AatII sites was performed using forward primer 5'-TTCTTAATTAAGGCGGGCGTTGCCAGGCGGGTGGC-3' and reverse primer 5'-ACTGACGTCCTCTATAGACACTGGCCAAGTCTGAGGGAGTC-3'. Amplification of the $Lkb1$ left homology arm and the addition of flanking 5' FseI and 3' AscI sites as performed using forward primer 5'-ATAGGCGCGCCAGCTGCTCTTATTTTGCACAGGAAACGTG-3' and reverse primer 5'-ATAGGCCGGCCAAAGAAGCCAGGCGCGACTTG-3'. The final targeting vector was then maxi-prepped and linearized with PmeI prior to purification by phenol/chloroform extraction.

**Generation of $Lkb1^{XTR}$ allele**. The linearized targeting plasmid was electroporated into 129-derived ES cells using standard conditions. Neomycin-resistant colonies were picked, expanded, and screened for successful targeting by PCR (Left-Arm Junction: forward primer 5'-AGCACTTTTCCCACCTTTCC-3' and reverse primer 5'-GGGGGAACTTCCTGACTAGG-3'; Right-Arm Junction: forward primer 5'-TGGCACAAAGCTTAGCCATA-3' and reverse primer 5'-GCCTGGCTCATTTCTGTGTT-3'). Of 282 clones, one (0.35%) was successfully targeted. Blastocysts were injected with targeted ES cells, yielding four high-percentage male chimeras. A germline chimeric $Lkb1^{XTR(neo)/+}$ male was crossed with $Rosa26^{FLPe}$ mice (The Jackson Laboratory: stock no. 003946) and the progeny were screened for NeoR deletion (forward primer 5'-CTACCCCCATCTATCCCTGAGCGTCACC-3' and reverse primer 5'-CGTTGGCCCGTGGGGACTCTTTATCG-3') and retention of the intact $Lkb1^{XTR}$ allele (forward primer 5'-CCCTCTTTGGGCCAGGTC-3' and reverse primer 5'-CCCCCTGAACCTGAAACATA-3'). $Lkb1^{XTR/+}$; $Rosa26^{FLPe/+}$ mice were crossed with wild-type 129 mice and their progeny were screened for loss of $Rosa26^{FLPe}$, thereby isolating $Lkb1^{XTR/+}$ mice for intercrossing to yield $Lkb1^{XTR/XTR}$ mice (The Jackson Laboratory: stock no. 034052)[108]. $Lkb1^{TR/+}$ mice were generated by crossing $Lkb1^{XTR/XTR}$ mice to CMV-Cre mice (The Jackson Laboratory: stock no. 006054)[41]. $Lkb1$ wild-type and $Lkb1^{XTR/+}$ mouse embryonic fibroblasts (MEFs) were isolated from embryos between E12.5 and E16.5 prior to transduction with either Adeno-Cre and/or Adeno-FLPo, which were obtained from the Gene Transfer Vector Core at the University of Iowa.

**Generation of lentiviral vectors**. Lenti-sgRNA/Cre vectors encoding individual sgRNAs were generated as previously described[109]. Lenti-sgRNA/Cre vectors encoding two or three tandem sgRNA cassettes were constructed as described previously[36]. Briefly, individual sgRNAs were cloned into plasmids encoding mU6, hU6, or bU6 promoters and unique constant regions (Addgene plasmids #85995, 85996, and 85997) by site-directed mutagenesis. The resulting U6-sgRNA cassettes were amplified and appended with flanking homology sequences to enable concatenation within the pLL3.3 Lenti-Cre backbone (previously linearized by PCR) by Gibson assembly. The final sgRNA sequences cloned into lentiviral vectors are listed in Supplementary Table 1. The primer sequences used for cloning sgRNAs and assembling multi-sgRNA vectors are listed in Supplementary Table 2. The Neo1, Neo2, Neo3, non-targeting (NT)1, NT2, $Lkb1$, $Sik1$, $Sik3$ sgRNA sequences have been previously described[36,42,109].

Lenti-sgRNA/Cre vectors were then diversified via the addition of sgID-BC cassettes as described previously to enable multiplexing[110]. In brief, unique sgID-BC inserts flanked by BamHI and BspEI sites were produced via PCR with Lenti-sgRNA/Cre as a template using unique forward primers encoding the sgID-BC region and a universal reverse primer. The sgID-BC amplicons were then digested with BamHI and BspEI and ligated into the Lenti-sgRNA/Cre backbones that had been previously linearized using BamHI and XmaI. The resultant colonies were then pooled for each vector prior to plasmid DNA extraction.

To generate lentivirus, Lenti-sgRNA/Cre vectors were individually co-transfected into 293T cells using polyethylenimine along with pCMV-VSV-G (Addgene #8454) envelope and pCMV-dR8.2 dvpr (Addgene #8455) packaging plasmids. Viral supernatants were collected at 36 and 48 hours post-transfection, passed through a 0.45-μm filter (Millipore: SLHP033RB), and sedimented by ultracentrifugation ($1.12 \times 10^5 \times g$ for 1.5 h at 4 °C), prior to resuspension in sterile PBS overnight at 4°C. Each virus was titered against a lentiviral Cre stock of known titer using immortalized LSL-YFP MEFs (Dr. Alejandro Sweet-Cordero/UCSF). Each lentivirus was stored at −80 °C and later thawed and diluted or pooled at equal ratios for multiplexed experiments prior to use in vivo.

**Mice, tumor initiation, and treatment**. $Kras^{LSL-G12D}$ (The Jackson Laboratory: stock no. 008179), $p53^{flox}$ (The Jackson Laboratory: stock no. 08462), $H11^{LSL-Cas9}$ (The Jackson Laboratory: stock no. 027632), $Rosa26^{FLPo-ERT2}$ (The Jackson Laboratory: stock no. 018906), and $Rosa26^{LSL-tdTomato}$ (ai9 and ai14 alleles; The Jackson Laboratory: stock no. 007909 & 007908) mice have been previously described[109,111–114]. All mice were on a C57BL/6J:129 mixed background except for NOD/SCID/γc (NSG; The Jackson Laboratory: stock no. 005557) mice used for transplantation experiments. The $Rosa26^{LSL-tdTomato}$ ai14 allele was implemented specifically with the $Lkb1^{XTR}$ mice, as the ai9 allele retains an additional FRT site within the PGK-NeoR cassette, which renders the tdTomato coding sequence susceptible to FLPo-ERT2-mediated deletion[112]. All mouse experiments included cohorts of mixed male and female mice aged 6 to 12 weeks (at tumor initiation) for autochthonous lung tumor models and 6 to 10 weeks for allograft models.

Lung tumors were initiated via intratracheal delivery of 60 μL of lentiviral Cre diluted in sterile PBS[115]. For comparing lung tumor burden between $Lkb1$ wild-type and $Lkb1^{XTR/XTR}$ contexts, tumors were initiated in KT, KT; $Lkb1^{XTR/XTR}$ mice with $7.50 \times 10^4$ IFU Lenti-Cre. To assess the impact of long-term Lkb1 restoration on tumor burden, KT, KT; $Lkb1^{XTR/XTR}$, and KT; $Lkb1^{XTR/XTR}$; FLPo-$ER^{T2}$ mice were transduced with $7.50 \times 10^4$ IFU Lenti-Cre. For survival analysis, lung tumors were initiated in KT, KT; $Lkb1^{XTR/XTR}$, and KT; $Lkb1^{XTR/XTR}$; FLPo-$ER^{T2}$ mice with $2.50 \times 10^5$ IFU Lenti-Cre. To generate tumors for longitudinal μCT and $^{18}$F-FDG PET/CT imaging, KT; $Lkb1^{XTR/XTR}$, and KT; $Lkb1^{XTR/XTR}$; FLPo-$ER^{T2}$ mice were transduced at $1.50 \times 10^4$ IFU/mouse. To assess the acute response to Lkb1 restoration at the histological, transcriptional (bulk and single-cell), and proteomic levels, tumors were initiated in KT; $Lkb1^{XTR/XTR}$, and KT; $Lkb1^{XTR/XTR}$; FLPo-$ER^{T2}$ mice using $7.50 \times 10^4$ IFU Lenti-Cre. For bulk RNA-seq, $Lkb1$ wild-type tumors were generated via transduction of KT mice with $2.50 \times 10^4$ IFU/mouse. For both primary and secondary Tuba-seq screens, KT and KT;$H11^{LSL-Cas9}$ mice were transduced with $2.50 \times 10^5$ and $1.00 \times 10^5$ IFU pooled Lenti-sgRNA/Cre, respectively. To validate tumor-suppressive capacity C/EBP transcription factors, tumors were initiated in KT and KT;$H11^{LSL-Cas9}$ mice using $5.00 \times 10^4$ IFU of either Lenti-sgInert/Cre (sgNeo1/sgNT1/sgNeo2) or Lenti-sgCebps/Cre (sg$Cebpa$/sg$Cebpb$/sg$Cebpd$). Finally, to generate tumors for comparing the gene expression profiles of C/ebp- and $Lkb1$-targeted tumors, KT;$H11^{LSL-Cas9}$ mice were transduced with either Lenti-sgInert/Cre, Lenti-sgCebps/Cre, or Lenti-sg$Lkb1$/Cre at $7.50 \times 10^4$ IFU/mouse.

Mice were administered tamoxifen (Sigma-Aldrich: T5648) in doses of 4 mg as indicated for each experiment. In general, mice received single doses on 2 consecutive days, followed by weekly single doses for the duration of the experiment. Tamoxifen was dissolved in a mixture of 10% ethanol (Sigma-Aldrich: E7023) and 90% corn oil (Sigma-Aldrich: C8267) to a concentration of 20 mg/mL and delivered via oral gavage. For measurement of BrdU incorporation, mice were administered 50 mg/kg BrdU (BD Pharmingen: 557892) intraperitoneally at 24 h prior to tissue harvest. BrdU was resuspended in sterile PBS to a concentration of 10 mg/mL. Mice were housed at 22 °C ambient temperature with 40% humidity and a 12-h light/dark cycle. The Stanford Institute of Medicine Animal Care and Use Committee approved all animal studies and procedures

**qRT-PCR**. Tissues for assessing $Lkb1$ mRNA levels were flash-frozen immediately following harvest. While thawing in preparation for lysis, tissues were manually disrupted on dry ice using RNase-Free Disposable Pellet Pestles (Thermo Fisher Scientific:12-141-368). Tissues were repeatedly passed through a 20 G needle to yield a finer homogenate prior to the addition of RLT buffer containing 1% β-mercaptoethanol. RNA was extracted using Allprep DNA/RNA Mini Kit (Qiagen: 80204). cDNA was generated using the ProtoScript® First Strand cDNA Synthesis Kit (NEB: E6300). Measurements of $Lkb1$ and $Gapdh$ expression levels were performed in triplicate using gene-specific primers ($Lkb1$ Fwd: 5'-CGAGGGATGTTGGAGTATGAG-3'; $Lkb1$ Rvs: 5'-AGCCAGAGGGTGTTTCTTC-3'; $Gapdh$ Fwd: 5'-CAGCCTCGTCCCGTAGAC-3'; $Gapdh$ Rvs: 5'-CATTGCTGACAATCTTGAGTGA-3') and PowerUp™ SYBR™ Green Master Mix (Thermo Fisher Scientific: A25776) on an HT7900 Fast Real-Time PCR System with 384-Well Block Module (Applied Bioscience). Data were acquired using the Sequence Detection Systems Software v2.4.1 in Absolute Quantitation mode.

**Western blotting**. Pellets of sorted neoplastic cells were stored at -80°C and later lysed directly in NuPAGE™ LDS Sample Buffer (Thermo Fisher Scientific: NP0007) containing 5% β-mercaptoethanol (Sigma-Aldrich: M3148). Tissues for assessing LKB1 protein levels were flash-frozen immediately following harvest. While thawing in preparation for lysis, tissues were manually disrupted on dry ice using RNase-Free Disposable Pellet Pestles (Thermo Fisher Scientific:12-141-368). Tissues were repeatedly passed through a 20 G needle to yield a finer homogenate prior to the addition of RIPA buffer (Thermo Fisher Scientific: 89900) containing proteinase/phosphatase inhibitor cocktail (Thermo Fisher Scientific: 78442). For bulk tissue lysates, protein concentration was measured using BCA protein assay kit (Thermo Fisher Scientific: 23250). For sorted cells, a fixed number of cells was loaded into each well, whereas for bulk lysates, 25 μg of lysate was loaded into each well of 4-12% Bis-Tris gels (Thermo Fisher Scientific: NP0323). Electrophoresis was performed with MES buffer (Thermo Fisher Scientific: NP0002) and resolved lysates were subsequently transferred to polyvinyl difluoride (PVDF) membranes (BioRad: 162-0177) according to standard protocols. Membranes were blocked in 5% milk and subsequently probed with primary antibodies against LKB1 (Cell Signaling Technology: 13031; 1:1000 dilution) and GAPDH (Cell Signaling Technology: 5174; 1:10,000 dilution), as well as secondary HRP-conjugated anti-mouse (Santa Cruz Biotechnology: sc-2005) and anti-rabbit (Santa Cruz Biotechnology: sc-2004) antibodies. Blots were visualized using Supersignal® West Dura Extended Duration Chemiluminescent Substrate (Thermo Fisher Scientific: 37071) and exposed on blue autoradiography film (Morganville Scientific: FM0200).

**Histology and immunohistochemistry**. Lung lobes were fixed in 4% formalin for 24 h, stored in 70% ethanol, and later paraffin embedded. Hematoxylin–eosin staining was performed using standard methods. Total tumor burden (tumor area/total area x 100%) and individual tumor sizes were calculated using ImageJ. Immunohistochemistry was performed on 4-μm sections using the Avidin/Biotin Blocking Kit (Vector Laboratories: SP-2001), Avidin-Biotin Complex kit (Vector Laboratories: PK-4001), and DAB Peroxidase Substrate Kit (Vector Laboratories: SK-4100) following standard protocols using the Sequenza system. The following primary antibodies were used: Cleaved Caspase 3 (Cell Signaling Technologies: 9661S; 1:100 dilution), phosphorylated Histone H3 Serine 10 (Cell Signaling Technologies: 9701S; 1:100 dilution), Ki-67 (BD Biosciences: 550609; 1:100 dilution), NKX2-1 (Abcam: ab76013; 1:200 dilution), and HMGA2 (Biocheck: 59170AP; 1:1000 dilution). For, Ki-67 staining, the mouse-on-mouse immunodetection kit (Vector Laboratories: BMK-2202) was used to block endogenous mouse IgG. IHC was performed using Avidin/Biotin Blocking Kit (Vector Laboratories: SP-2001), Avidin-Biotin Complex kit (Vector Laboratories: PK-4001), and DAB Peroxidase Substrate Kit (Vector Laboratories, SK-4100) following standard protocols. Sections were developed with DAB and counterstained with hematoxylin. The frequency of H3P- and Ki-67-positive nuclei were quantified using ImageJ on images of ×20 fields, and cleaved caspase 3-positive cells were quantified by direct counting on images of ×20 fields.

**Mouse cell lines**. Cell lines were generated from primary tumors from $Kras^{LSL-G12D};Trp53^{flox/flox}$ (cell line 394T4[44]), $Kras^{LSL-G12D};Trp53^{flox/flox};Lkb1^{XTR/XTR};Rosa26^{LSL-tdTomato}$ (cell line 3406) and $Kras^{LSL-G12D};Trp53^{flox/flox};Lkb1^{XTR/XTR};Rosa26^{FlpOER/LSL-tdTomato}$ (cell lines 2841T6, 3841T4, and 2804T5B) mice previously transduced with lentiviral Cre. To establish cell lines, individual tumors were micro-dissected from tumor-bearing lungs, minced, and directly cultured in DMEM (Thermo Fisher Scientific 11995081) supplemented with 10% FBS (Phoenix Scientific), 1% penicillin-streptomycin-glutamate (Thermo Fisher Scientific 10378016), and 0.1% amphotericin B (Thermo Fisher Scientific 15290018) at 37°C and 5% $CO_2$ until cell lines were established. Cells were authenticated for genotype. To induce $Lkb1$ restoration, cells were treated with either 1 μM 4-hydroxytamoxifen (4-OHT; Sigma Aldrich H7904) dissolved in 100% ethanol or vehicle (1:2000 100% ethanol). All cell lines included in this study tested negative for mycoplasma contamination (Lonza MycoAlert Mycoplasma Detection Kit, #LT07-218).

**Cell cycle and cell death assays**. For cell cycle and death analyses, $2.00 \times 10^5$ cells were seeded per well within 6-well plates in triplicate for each experimental group. Cells were treated with 4-OHT or ethanol for 48 h prior to re-plating at $1.00 \times 10^5$ cells per well. Culture media was changed at 72 h. At 96 h, the cells were 30–50% confluent, when the culture media containing detached dead cells were collected and later combined with trypsinized, detached cells. The attached cells were labeled with EdU for 45 min, washed, trypsinized, and counted. $1.00 \times 10^5$ cells were subjected to EdU-AF647 (Thermo Fisher Scientific C10424) and FxCycle Violet (Thermo Fisher Scientific F10347) double staining following the vendor's protocols. Another $1.00 \times 10^5$ cells were stained with Annexin V (Thermo Fisher, A23204) and DAPI. $5.00 \times 10^4$ events were recorded on a BD LSRII flow cytometer, gated on forward/side scatter and singlets, and analyzed for cell cycle or death. Representative gating schemes for cell cycle and cell death analyses are included in Supplementary Fig. 19a, b.

**μCT and ¹⁸F-FDG PET/CT imaging**. Serial measurements of tumor size were captured by μCT using the Trifoil CT eXplore CT120 with respiratory gating. Using MicroView v.2.5.0 (Parallax Innovations), lung tumors were captured within advanced regions generated using the spline function and measures of tumor volume were acquired by determining the volume of voxels that fall within a defined range of pixel intensity that corresponds to each tumor mass. Measures of tumor size were reported in terms of size relative to the initial measurement of a given tumor. Mouse imaging was staggered on the basis of initial tumor detection and thus the relative timing of tumor measurements was variable. To align measurement intervals across all tumors in the study, interpolated values were used to aggregate tumors into cohorts, and measures of tumor size were reported in terms of weeks relative to treatment initiation. Non-interpolated measurements are shown in Supplementary Fig. 6b.

¹⁸F-FDG-PET/CT imaging was performed as previously described[116,117]. ¹⁸F-FDG tracer was delivered by retro-orbital injection. ¹⁸F-FDG signal was measured in terms of maximum percent injected dose per gram (%ID/g) for each tumor. Measurements of ¹⁸F-FDG uptake were reported in terms of fold change relative to pre-treatment levels.

**Tumor dissociation and cell sorting**. Micro-dissected lung tumors were dissociated with collagenase IV, dispase, and trypsin at 37°C for 30 minutes as previously described[118]. The digestion buffer was then neutralized with cold L-15 media (Thermo Fisher Scientific: 21083027) containing 5% FBS (Gemini Bio) and DNaseI (Sigma-Aldrich: DN25). Dissociated cells were treated with ACK Lysis Buffer (Thermo Fisher Scientific: A1049201) and resuspended in PBS containing 2 mM EDTA (Promega: V4233), 2% FBS, and DNase I. For the isolation of neoplastic cells, dissociated cells were stained with DAPI and antibodies against CD45 (BioLegend: 103112; 1:800 dilution), CD31 (BioLegend: 102402; 1:800 dilution), F4/80 (BioLegend: 123116; 1:800 dilution), and Ter119 (BioLegend: 116212; 1:800 dilution) to exclude hematopoietic and endothelial cells. FACSAria™ sorters (BD Biosciences) were used for cell sorting. For sorting of total cells within tumors for single-cell RNA-seq, dissociated cells were stained with DAPI only to exclude dead cells. Representative gating scheme included in Supplementary Fig. 20a.

**Intratracheal transplant of neoplastic cells**. For the transplant of treatment-naïve neoplastic cells from $KT; Lkb1^{XTR/XTR};FLPo-ER^{T2}$ donor mice, tumor-bearing lungs were extracted en bloc, dissociated into individual lobes, and maintained on ice. Individual tumor nodules were then extracted under a dissecting microscope, minced, and aggregated prior to enzymatic dissociation as described in the Tumor Dissociation and Cell Sorting section. Following red blood cell lysis and resuspension in PBS, a minor fraction of the resulting single-cell suspension was set aside for staining and flow cytometric analysis as described in the Tumor Dissociation and Cell Sorting section to determine the density of lineage-negative, tdTomato^positive cells. The remaining neoplastic cells suspended in PBS were administered to NSG recipient mice via intratracheal delivery. For the transplant of neoplastic cells derived from tamoxifen-treated $KT;Lkb1^{XTR/XTR}$, and $KT; Lkb1^{XTR/XTR};FLPo-ER^{T2}$ donor mice, the density of neoplastic cells within each donor suspension was first assessed by flow cytometry and then normalized such that each NSG recipient cohort received an equal number of neoplastic cells ($5.00 \times 10^4$ cells/mouse).

**Bulk RNA-seq library preparation**. Total RNA was prepared from sorted pellets of neoplastic cells ranging from $2.50 \times 10^4$ to $1.00 \times 10^5$ cells using the AllPrep DNA/RNA Micro Kit (Qiagen: 80284). RNA quality was assessed using the RNA6000 PicoAssay kit on the Agilent Bioanalyzer 2100, and samples with a RIN score below seven were excluded. Two to ten nanograms of total RNA served as input for the preparation of RNA-Seq libraries using the Trio RNA-Seq, Mouse rRNA kit (Tecan Genomics: 0507-32). Purified libraries were assessed using the Agilent High Sensitivity DNA kit (Agilent Technologies: 5067-4626) and then sequenced on an Illumina NextSeq 550 (2 × 75 bp High-Output).

**Analysis of bulk RNA-seq data**. Paired-end bulk RNA-seq reads were aligned to the mm10 mouse genome using STAR (v2.6.1d) 2-pass mapping with standard parameters and an sjdbOverhang of 75 bp. Estimates of transcript abundance were obtained using RSEM (v1.2.30) using standard parameters[119,120]. The differentially expressed genes between different tumor genotypes were called by DESeq2 (v1.26.0) using transcript abundance estimates via tximport[121,122]. The DESeq2-calculated fold changes were used to generate ranked gene lists for input into GSEA (v3.0)[123]. GSEA results using the GO Biological Process module were imported into Cytoscape (v3.8.2) with the EnrichmentMap plugin for network construction using default parameters[124,125]. Networks were ported to R using ggraph (v2.0.4) and clusters of related GO terms were defined using the edge betweenness community detection algorithm in igraph (v1.2.6)[126]. K-means clusters were defined in ComplexHeatmap (v2.2.0) and GO term enrichment analysis was performed using compareCluster in ClusterProfiler (v3.14.3)[127,128]. For motif enrichment, the differentially expressed genes with absolute log₂ fold changes >1 and a false discovery rate <0.05 were compiled into gene lists, converted to RefSeq identifiers using biomaRt, and used as input for Pscan (−450 to +50 bp from the TSS) using either the JASPAR 2018 non-redundant or TRANSFAC databases[129].

**Analysis of previously published gene expression datasets.** Gene expression data derived from lung tumors in genetically engineered mice under accession numbers GSE6135, GSE21581, GSE69552, and GSE133714 were acquired from NCBI Gene Expression Omnibus using the GEOquery package[25,52,53,130]. Differential expression was computed using limma, and the resulting log₂ fold changes were used to generate ranked gene lists for input into GSEA[123,131]. For the comparison of $Kras^{G12D}$ and $Kras^{G12D};Nkx2-1^{\Delta/\Delta}$ tumors by RNA-seq, log₂ fold changes were directly downloaded from https://doi.org/10.7554/eLife.38579.019[86].

For the generation of signatures of alveolar type I and type II identities, single-cell gene expression data were acquired from the sources below[54,66–72]. Each dataset was loaded into Seurat and the FindMarkers function (only.pos = TRUE, min.pct = 0.25, logfc.threshold = 0.25) was performed on the basis of the curated cell type identities for each dataset. The cut-offs used for each dataset to establish gene sets are listed. Gene sets were then compiled into a GMT file using GSEABase (v1.48.0).

Tabula Muris/Tabula Muris Senis[67,71]—processed Seurat objects obtained from https://www.synapse.org/#!Synapse:syn21560554

Mouse Cell Atlas[72]—fetal and adult lung DGE matrices accessed at http://bis.zju.edu.cn/MCA/dpline.html?tissue=Lung

Strunz, et al.[66]—lung epithelial high-resolution Seurat object obtained from https://github.com/theislab/2019_Strunz

Little, et al.[68]—Cell Ranger outputs for control lung obtained from the Gene Expression Omnibus: GSE129584

Angelidis, et al.[69]—Seurat object obtained from https://github.com/gtsitsiridis/lung_aging_atlas

Guo, et al.[70]—DGE matrices derived from developing lungs obtained from the Gene Expression Omnibus: GSE122332

Treutlein, et al.[54]—processed SingleCellExperiment file obtained courtesy of the Hemberg Lab at https://hemberg-lab.github.io/scRNA.seq.datasets/mouse/tissues/

**Mass spectrometry.** Pellets of sorted neoplastic cells were stored at −80 °C and later resuspended in PBS prior to loading on a Whatman QM-A Quartz Microfiber Filter (GE Life Science: 1851-047) microreactor tip. Samples were then lysed, digested with trypsin/lys-c (Promega: V5073), and de-salted using C18 (Empore: 320907D) stage tips as described previously[132]. Eluted samples were dried and resuspended with Solution A (2% ACN, 0.1% FA) for mass spectrometry analysis.

Samples were analyzed on the timsTOF Pro (Bruker Daltonics), an ion-mobility spectrometry quadrupole time of flight mass spectrometer. Specifically, a nanoElute (Bruker Daltonics) high-pressure nanoflow system was connected to the timsTOF Pro. Peptides were delivered to reversed-phase analytical columns (25 cm × 75 μm i.d., Ionopticks: AUR2-25075C18A-CSI). Liquid chromatography was performed at 40 °C, and peptides were separated on the analytical column using a 120 min gradient (solvent A: 2% ACN, 0.1% FA; solvent B: 0.1% FA, in ACN) at a flow rate of 400 nL/min. A linear gradient was applied for 60 min to 15%, 30 min to 23%, 10 min to 35%, followed by a step to 80% B in 10 min and held for 10 min for wash. The timsTOF Pro was operated in PASEF mode with the following settings: Mass Range 100 to 1700m/z, 1/K0 Start 0.60 V·s/cm², End 1.6 V·s/cm², Ramp time 100 ms, Lock Duty Cycle to 100%, Capillary Voltage 1600, Dry Gas 3 l/min, Dry Temp 180 °C, PASEF settings: 10 MS/MS, Scheduling Target intensity 500000, CID collision energy 10 eV.

Bruker raw data files were analyzed by msfragger using the tool FragPipe[133]. Msfragger was run using the default modifications with an error tolerance of 20 ppm for precursors and + / − 40 ppm for fragments. We used a mouse protein database downloaded from RefSeq on 06/18/2018. Peptide and protein identifications were validated using PeptideProphet and quantitation was done using IonQuant with 'match between runs' and selecting the MaxLFQ method.

Razor intensities were analyzed using the DEP package[134]. Briefly, contaminants and species detected in less than 25% of samples were filtered out. Intensities were normalized by variance stabilizing transformation and missing values were imputed using the MinProb imputation method. Differential expression was assessed using the limma package[131]. Fold changes calculated with limma were used to generate ranked gene lists for input into GSEA[123]. GSEA results using the GO Biological Process module were imported into Cytoscape with the EnrichmentMap plugin for network construction using default parameters[124,125]. Networks were ported to R using ggraph and clusters of related GO terms were defined using the edge betweenness community detection algorithm in the igraph package[126].

**Single-cell RNA-seq library preparation.** Sorted cells were pelleted at 300 × g for 5 min to resuspend in PBS. Cell density was then assessed on a hemacytometer and adjusted to the target density. Cells were loaded in each channel of Chromium Chip B (10X Genomics: 1000074) with a recovery target of 8,000 cells per sample, and emulsions were generated on the Chromium Controller (10X Genomics). Libraries were constructed using the Chromium Single Cell 3′ Library & Gel Bead Kit v3 kit (10× Genomics: 1000075). 10× libraries derived from total cells were sequenced on Illumina NextSeq 500 and Hi-Seq 2500 platforms (26 bases for Read 1, 8 bases for i7 Index 1, and 91 bases for Read 2), whereas sorted neoplastic cell libraries were sequenced on an Illumina Nova-seq 6000 (26 bases for Read 1, 8 bases for i7 Index 1, and 90 bases for Read 2).

**Analysis of single-cell RNA-seq data.** Reads were aligned to the mm10 genome and feature counts were obtained using Cell Ranger (v3.0.2) (10× Genomics). Feature-barcode matrices were then imported into R using Seurat (v3.2.0) (minimum of 500 features/cell for sorted neoplastic dataset and 200 features/cell for the total cell dataset) and merged into Seurat objects for pre-processing, normalization (regressing out nCount_RNA in ScaleData), dimensional reduction (2000 variable features for each dataset, 6 and 25 dimensions were used for sorted neoplastic and total cell datasets, respectively), and clustering (resolutions of 0.3 and 0.1 were passed to FindClusters to implement the Louvain algorithm for community detection within sorted neoplastic and total cell datasets, respectively)[135]. Cells were filtered on the basis of percent mitochondrial reads and maximum feature count using percentile-based cutoffs. For the sorted neoplastic cell dataset, putative stromal cells ($Pecam1^+$, $Cdh5^+$, $Ramp2^+$ endothelial cells and $Col1a1^+$, $Col1a2^+$, $Col3a1^+$, $Mgp^+$ fibroblasts) were filtered out following preliminary clustering analysis. For the total cell dataset, tdTomato$^{positive}$ cells outside of the epithelial compartment were excluded from downstream analyses. For cell-type prediction within the total cell dataset, SingleR (v1.4.1) was used with the Tabula Muris lung dataset as a reference (following conversion of Seurat object to SingleCellExperiment and transformation with logNormCounts in scater)[71,136,137]. For the total cell dataset, Louvain-based clusters were collapsed into pseudobulk samples and differential expression analysis between restored and non-restored samples was performed using muscat (v1.4.0)[138].

Trajectory inference analysis was performed using Monocle3 (v0.2.1) with standard parameters (following conversion to CellDataSet; close_loop set to FALSE in learn_graph; ATII-like subpopulations was manually defined as the root population)[139]. For RNA velocity analysis, spliced, unspliced, and ambiguous matrices were generated using the run10x command in velocyto (v0.17.17) with default parameters[140]. The resulting loom files were imported into Seurat with the ReadVelocity command in SeuratDisk, integrated with meta data, subsetted to only those cells that passed QC before, and exported to h5ad format. RNA velocity analysis was then performed using scvelo (v.0.2.3) with standard parameters[141].

**Isolation of genomic DNA from mouse lungs and preparation of Tuba-seq libraries.** Genomic DNA was isolated from bulk tumor-bearing lung from each mouse following the addition of three spike-in controls (5.00 × 10⁵ cells per control) to enable absolute quantification of cell number using Tuba-seq as described previously[110]. Libraries were prepared by single-step amplification of the sgID-BC region from a total of 32 μg of genomic DNA per mouse across eight 100-μL reactions using NEBNext Ultra II Q5 Master Mix (New England Biolabs: M0544L). To enable computational removal of chimeric reads that result from index hopping during ultra-deep sequencing, the sgID-BCs were amplified using defined dual-indexing primer pairs with unique i5 and i7 indices. The unique dual-indexed primers used were forward: AATGA-TACGGCGAC

CACCGAGATCTACAC- 8-nucleotide i5 index -ACACTCTTTCCCTACACGACGC TCTTCCGATCT-6N to 9 N (random nucleotides to increase diversity)-GCGCACGT CTGCCGCGCTG and reverse: CAAGCAGAAGACGGCATACGAGAT- 6-nucleotide i7 index -GTGACTGGAGTTCAGACGTGTGCTCTTCCGATCT-6N to 9N (random nucleotides to increase diversity) -CAGGTTCTTGCGAACCTCAT. The PCR products were subjected to double-sided purification using Agencourt AMPure XP beads (Beckman Coulter: A63881). Purified libraries were assessed using the Agilent High Sensitivity DNA kit (Agilent Technologies: 5067-4626) on the Agilent 2100 Bioanalyzer (Agilent Technologies: G2939BA). Individual libraries were pooled in a weighted format on the basis of total lung weight, and the final pool was cleaned up using a single-sided purification with Agencourt AMPure XP beads. Libraries were sequenced on Illumina® HiSeq 2500 and NextSeq 500 platforms to obtain 150-bp paired-end reads.

**Tumor barcode sequencing analysis.** Only those reads containing complete sgID-BC cassettes (8-nucleotide sgID region and 30-nucleotide barcode: GCNNNNNTANNNNNGCNNNNNTANNNNNGC) were retained. Each sgID corresponds to a unique Lenti-sgRNA/Cre vector included in the lentiviral pool, whereas the 20N random nucleotide basis serves as a unique clonal identifier for each tumor. sgIDs were designed with a minimum hamming distance of three nucleotides. Read pairs exhibiting mismatches within this sgID-BC region were discarded to minimize the impact of sequencing error. Furthermore, we required perfect matching between sgIDs in forward reads with the known sgIDs that were included within each pool. Reads were then aggregated on the basis of the random barcode region to create unique barcode pileups that represent individual tumors. Tumors with random barcodes containing indels were discarded to avoid potential alignment errors and miscalculation of distances between barcodes. Any tumor with a barcode within a hamming distance of two nucleotides from a larger tumor was considered spurious and excluded to minimize the impact of PCR and sequencing errors. Measures of absolute cell number for each tumor were then calculated by multiplying the read counts for each barcode pileup (tumor) by the size of the spike-in controls (1.00 × 10⁵ cells) and subsequently dividing by the average number of reads within each mouse for the three barcodes corresponding to the three spike-in controls that were added in during tissue processing. For the primary Tuba-seq screen, the median sequencing depth was ~1 read per 15 cells, and the minimum sequencing depth is ~1 read per 170 cells. For the secondary, C/ebp-targeted Tuba-seq screen, the median sequencing depth was ~1 read per 8 cells, and the minimum sequencing depth is ~1 read per 17 cells. The impact of GC

amplification bias on tumor size was accounted for as described previously[42]. Tumor size cut-offs of 50 and 100 cells were applied for the primary and secondary Tuba-seq screens, respectively.

Multiple metrics of tumor size distribution were examined, including various percentiles as well as the maximum-likelihood estimate of the mean assuming a log-normal distribution of tumor size[42]. Confidence intervals and $p$ values were calculated by a nested bootstrap resampling approach to account for variation in sizes of tumors of a given genotype both across and within mice. First, the tumors of each mouse were grouped, and these groups (mice) were resampled. Second, all tumors of a given mouse resampling were bootstrapped on an individual basis (500 repetitions). False discovery rates were calculated using the Benjamini-Hochberg procedure.

**Analysis of previously published NKX2-1 ChIP-seq data**. BedGraph files under accession GSM1059354 corresponding to NKX2-1 ChIP-seq performed on oncogenic KRAS-driven lung tumors were acquired from NCBI Gene Expression Omnibus[85]. De novo motif enrichment at NKX2-1 bound sites was performed using HOMER findmotifs.pl using default parameters[142]. Gene associations to NKX2-1-bound sites were generated using GREAT analysis using mm9 assembly and a window of $-2$ to $+1$ kb relative to TSS[143].

**Reporting summary**. Further information on research design is available in the Nature Research Reporting Summary linked to this article.

## Data availability

Next-generation sequencing data for the Tuba-Seq and RNA-Seq (bulk and single-cell) experiments are accessioned under the GSE179560 SuperSeries at NCBI Gene Expression Omnibus. Shotgun proteomics data are accessioned under PXD026738 at PRIDE. Lenti-sgRNA/Cre plasmids generated in this study are available through the Winslow Lab plasmid collection on Addgene [https://www.addgene.org/Monte_Winslow/]. $Lkb1^{XTR/XTR}$ mice generated in this paper are available at The Jackson Laboratory (Stock no. 034052). $Lkb1^{XTR/XTR}$ mouse lung cancer cell lines are available from the corresponding author upon request. JASPAR 2018 non-redundant [https://jaspar.genereg.net/api/v1/live-api/] and TRANSFAC [http://cisbp.ccbr.utoronto.ca/index.php] databases are publicly available and accessible via Pscan [http://159.149.160.88/pscan/]. Previously published gene expression data derived from lung tumors in genetically engineered mice are available under accession numbers GSE6135, GSE21581, GSE69552, and GSE133714 at NCBI Gene Expression Omnibus. Lung cell identity gene expression signatures were derived from publicly available single-cell RNA-seq datasets, including Tabula Muris & Tabula Muris Senis [https://www.synapse.org/#!Synapse:syn21560554]; Mouse Cell Atlas [http://bis.zju.edu.cn/MCA/dpline.html?tissue=Lung]; Strunz et al.[66] [https://github.com/theislab/2019_Strunz]; Little et al.[68]—Gene Expression Omnibus: GSE129584; Angelidis et al.[69] [https://github.com/gtsitsiridis/lung_aging_atlas]; Guo et al.[70]—Gene Expression Omnibus: GSE122332; Treutlein, et al.[54] [https://hemberg-lab.github.io/scRNA.seq.datasets/mouse/tissues/]. Source data are provided with this paper.

## Code availability

All custom codes used in this work are available from the corresponding author upon request. Scripts for analyzing the Tuba-seq datasets are available at https://github.com/lichuan199010/Tuba-seq-analysis-and-summary-statistics.

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

## Acknowledgements

We thank the Stanford Shared FACS facility, the Stanford Center for Innovation in in vivo Imaging, the Stanford Functional Genomics Facility, the Stanford Transgenic, Knockout and Tumor Model Center, and Stanford Animal Histology Services for technical support; A. Orantes for administrative support; L. Penland, N. Neff, N. Hughes, and L. Cong for support with single-cell RNA-seq; M. Yousefi and R. Tang for help with FACS; L. Andrejka for help with generating Tuba-seq libraries. C. Li, E. Shuldiner, and D. Petrov for support with Tuba-seq analysis; E. Snyder, J. Lipsick, and members of the Winslow laboratory for helpful comments. C.W.M. was supported by the NSF Graduate Research Fellowship Program and an Anne T. and Robert M. Bass Stanford Graduate Fellowship. J.J.B. was supported by NIH F32-CA189659. S.E.P. was supported by an NSF Graduate Research Fellowship Award and the Tobacco-Related Diseases Research Program Predoctoral Fellowship Award. H.C. was supported by a Tobacco-Related Disease Research Program (TRDRP) Postdoctoral Fellowship (28FT-0019). R.C. was supported by NIH 5T32GM007276. D.B.S. was supported by NIH R01-CA208642, DOD LCRP W81XWH-18-1-0295, and funding from the Jonsson Comprehensive Cancer Center. This work was supported by NIH R01-CA175336 (to M.M.W.), NIH R01-CA207133 (to M.M.W.), and NIH R01-CA230919 (to M.M.W.). This work was supported by a National Cancer Institute Cancer Center Support Grant (P30CA124435). The content is solely the responsibility of the authors and does not necessarily represent the official views of the NCI.

## Author contributions

J.J.B. designed and generated the *Lkb1^XTR* allele under the supervision of D.M.F. and M.M.W. C.W.M. performed tumor burden experiments, survival analysis, μCT imaging, RNA-seq, and Tuba-seq experiments under the supervision of M.M.W. C.W.M. and M.K.T. performed immunohistochemistry. S.E.P. and M.K.T. performed qRT-PCR and western blot analysis. H.C. and S.E.P. sorted neoplastic cells. H.C. performed in vitro analysis of cell cycle and cell death. M.H. performed 18F-FDG PET/CT imaging under the supervision of D.B.S. R.C. and J.D. acquired and processed the mass spectrometry data under the supervision of P.K.J. C.W.M. and M.M.W. wrote the manuscript with contributions from all authors.

## Competing interests

M.M.W. is a founder of, and holds equity in, D2G Oncology, Inc. The authors declare no other competing interests.
