## [Peer Review File · Nature Communications]

LKB1 drives stasis and C/EBP-mediated reprogramming to an alveolar type II fate in lung cancerREVIEWER COMMENTS

Reviewer #1 (Remarks to the Author):

In this manuscript Murray et al develop a genetically engineered mouse model of Kras-Lkb1 mutant lung cancer in which they subsequently restore Lkb1 activity in established tumors. First, they demonstrate that restoration of Lkb1 over a 6 week period after establishing tumors for 6 weeks prevents further tumor growth. They then establish tumors for longer periods (18-22 weeks) and assess the more acute impact of Lkb1 restoration on tumor cell transcriptomic profiles after 2 weeks, identifying C/ebp transcription factor motifs and ATII gene signatures enriched in restored tumors in bulk analyses. They also perform single cell RNAseq which again shows enrichment for AT-like gene sets. To confirm a role for C/ebp in regulating this program, they also knockout C/ebp in Kras-Lkb1 tumors and also observe induction of ATII-like program. Finally they identify C/EBPalpha as a dominant factor co-operating in part with NKX2-1 to drive this program downstream of LKB1.

In general this is an extremely thorough, well done body of work. I have only a few minor suggestions, largely caveats of the work that should be mentioned in the discussion.

1. Fig 1 – while they observe tumor stasis, this does not differentiate between whether Lkb1 restoration causes cell cycle arrest/differentiation of all tumor cells, or whether it kills off a population of cells, selecting for those that end up with the ATII phenotype. Have they also stained for markers of apoptosis such as cleaved caspase 3, as this could support a selective process if positive? Unless they have strong evidence that this is not the case this caveat should be added to the discussion.

2. Figs 2/3 – again this set of experiments is very elegant and convincing. However, parallel studies of restoring Lkb1 in vitro in an established cell line in a time-controlled fashion would further help to address my question above, and whether the “acute” transcriptomic effects are direct, or a consequence of selection for a cell state that can tolerate Lkb1 reconstitution. In their prior study (Murray et al., Cancer Discov 2019) the authors examined gene sets in human LKB1 reconstituted cell lines implicating SIK as a downstream activity. Have they similarly examined C/EBPalpha/ATII gene sets in this human cell line data (with the caveat that, depending on the time point after LKB1 reconstitution, one may see different signatures).

3. The discussion feels a bit slanted against the role of canonical AMPK signaling and intent on proving the novel role for SIK kinases and the ATII differentiation program (model in 5H). However, I feel that it would be helpful to include discussion of the caveats above. Specifically, while everything may be direct, an alternate possibility is they are uncovering how KRAS-LKB1 mutant cells successfully adapt to restoration of AMPK. For example, activation of YAP transcriptional programs can compensate for oncogenic KRAS suppression in an adaptive fashion, which is not direct.

Reviewer #2 (Remarks to the Author):

Murray et. al. demonstrate that restoration of the tumor suppressor LKB1 drives tumor cell growth arrest and transcriptional alveolar differentiation in lung adenocarcinoma (LUAD) cells. LKB1/STK11 is a frequently mutated tumor suppressor gene in treatment refractory human LUADs and its functions remains a topic of debate. Major strengths of the manuscript include the use of an elegant strategy (previously reported by the authors) to generate a genetically engineered mouse model (GEMM) whereby LKB1 expression can be restored in established tumors that initially arose in situ (in combination with Kras or Kras/p53 mutations). While loss of LKB1 in GEMM of LUAD has been previously reported, this study is the first to rigorously examine the consequences of LKB1 restoration (in a specifically controlled and timed manner) in vivo. The contribution of alveolar transcriptional programs to LUAD progression and their epigenetic regulation by lineage transcription factors (including C/EBPalpha and NK2X.1) is a known concept. However, the causal link between LKB1 loss of function and these lineage specific programs has not been clearly

established. Therefore, on balance, the findings in Murray et al. are appropriate in terms of scope and novelty for Nature Communications.

One important weakness is that the investigators do not provide any data regarding the grade and/or lineage marker status of their bulk tumors in vivo at the various time points of their experiments. It is particularly relevant to more clearly describe this when LKB1 restoration is initiated for several reasons. First, it is known that during LUAD progression (using similar Kras/p53 GEMM models) different lineage factors can be activated or suppressed as compensatory mechanisms of tumor suppression, and their expression or activity during tumor stasis or proliferation may be stage specific. Second, these GEMMs are likely to progress and transdifferentiate in a heterogeneous manner. Third, assuming the feasibility of re-activating tumor suppressor pathways in human NSCLC (as suggested by the authors) knowing what grade or human equivalent stage of LUAD are most susceptible to LKB1 restoration would help the readers interpret the potential impact of the findings. It is also possible that some genetic or stage specific context will not be responsive to LKB1 restoration alone when a so -called intermediate AT2/AT1 like state (the presumed target of restored LKB1 activity) has already completely eroded (see additional comments regarding p53 below).

Specific comments, questions, and suggestions are as follows:

- Fig 4 and S3: It was not always clear in the legends how the sampling was done. For example, how many tissue sections were used to assess tumor area and how many mice were used to quantify tumor size distributions?
- Figure S5: It is somewhat misleading to state that the effect of LKB1 restoration is qualitatively independent of p53 without commenting on the quantitative differences; in the absence of p53, the effects on tumor stasis are clearly more muted. This is also reflected in the gene expression analysis (Fig. S11). Is this because the grade of the tumors in p53 null animals are higher (when LKB1 is initially restored) or is there a context dependent molecular explanation for this, given the reported reciprocal relationship between LKB1 and p53 (in LUAD)? A more accurate description and discussion of these results is warranted.
- Fig S7 and S8: Some of the major data should be included as main figures, (journal policies allowing). Statistical significance seems to be missing for S8 in particular.
- While much of the supplemental transcriptomic/proteomic analysis are rigorous, this data is descriptive and, in some cases, somewhat redundant (e.g., S9 and S10, S17); It would be useful if some of this processed data could be summarized and then presented fully as supplemental tables for exploration by the readers.
- Fig S15 legend and related text: It is recommended that the authors clarify in the text that the lineage specific proliferation status is inferred by analyzing cell cycle gene markers from single cell RNA-seq data. It also seems highly likely that LKB1 suppresses proliferation in lineages that are not AT2 like. The conclusion in the manuscript could be more justified if functional confirmation in tumor tissue was provided; otherwise, the stated conclusions should be more aligned with the nuanced and indirect data that is presented.

Reviewer #3 (Remarks to the Author):

LKB1 is among the most frequently altered tumor suppressors in lung adenocarcinoma. In this manuscript, Murray et al. develop an elegant system to test whether restoration of Lkb1 expression would limit tumor progression in a KrasG12D/Lkb1-LOF mouse lung tumor model. They validate a novel mouse model (Lkb1XTR), in which Cre-mediated Lkb1 loss of function can be complemented by tamoxifen-regulated FLPo-ER mediated restoration of Lkb1 expression to examine roles for Lkb1 loss in tumor progression. Using this model, the authors demonstrate that restoration of Lkb1 expression does indeed constrain tumorigenesis driven by the combination of

Kras/Lkb1 loss of function. Further, the authors identify targets of C/EBP transcription factors as key downstream targets of Lkb1 that regulate tumor progression. The authors also demonstrate that Lkb1 restoration drives transcriptional programs relating to alveolar type 2 epithelial cell function, suggesting that roles for Lkb1 in the regulation of alveolar epithelial cell differentiation plays a role in tumor suppression. Overall, experiments are well designed, data of high quality, and are generally supportive of conclusions.

Concerns:

- 1) Multiple publications have shown that inactivation of Lkb1 in Kras driven mouse lung tumor leads to adenosquamous transition, and emergence of diverse histological subtypes. However, this is not seen in this study. The authors should discuss this issue further to help reconcile their findings with the existing literature.
- 2) A caveat with the LkbXTR model is that "restoration" of Lkb1 occurs not only within tumors but presumably increased Lkb1 occurs in all cells undergoing tamoxifen regulated FLPO mediated recombination. Does this lead to changes in the molecular phenotype of non-tumorigenic cells? In general, additional discussion should be included that identifies caveats with the model that impact interpretation of data.
- 3) Authors claim that restoration of Lkb1 drives transcriptional programs relating to alveolar type 2 cell function. However, data in Figure 2C suggest that "Lkb1-restored" tumor cells acquire a molecular phenotype more like those of Kras-induced tumors. Direct comparisons to wildtype AT2 cells are not made. These comparisons would be helpful in assessing changes in molecular phenotype of Lkb1-sufficient, -deficient (KT) and "restored" tumor cells. These comparisons would also provide insights into the relationship between observations in this study with other related reports. Additional discussion would be helpful.
- 4) In discussion, authors state that the inability of LKB1 re-expression to "recover the innate immune components due to stable epigenetic repression ..." contrasts with their finding that Lkb1 restoration suppresses tumor progression. How do the authors reconcile these observations – this should be discussed?
- 5) The concluding sentence starting on line 529 addresses Lkb1-dependent changes in the proliferative activity of tumors. This seems somewhat disconnected from the remainder of the paragraph which focuses on the diversity of histological subtypes and how this may be mediated by altered expression of Nkx2.1 and downstream target genes. Perhaps this paragraph should be re-focused for clarity.
- 6) Figure 3F and S13. Representation of cell types within scRNA-Seq data can not be directly equated with their abundance within tumors.
- 7) In Supplemental Figure S4, the authors transplant tumor cells from KT;Lkb1XTR/XTR;FLPO-ERT2 to NSG mice, followed by treatment with either vehicle or tamoxifen. A significant reduction of tumorigenesis was observed among Lkb1-restored compared to Lkb1-LOF tumors. Have the authors made comparisons with KT transplantation to determine whether Lkb1 restoration is comparable to KT?
- 8) In Supplemental Figure S5, a KPT alone control will help with data interpretation.
- 9) Typographical error in Supplemental Figure S11D: label of lower panel should be Higher in Non-Restored tumors.

Response to Reviewers
NCOMMS-21-24931-T

We thank the Reviewers for their careful assessment and critique of our manuscript, which has led to significant strengthening of this work. Our revised manuscript includes multiple new experiments in addition to modifications to the main text and figure legends at the request of the Reviewers. Please find our detailed responses to their comments shown in blue text below. Callouts to additions and/or modifications of text and figures are emboldened and underlined in blue text.

Major modifications made to the manuscript in order to address the Reviewers' comments are summarized below.

1. Complementary *Lkb1* restoration experiments *in vitro* to capture the acute effects of LKB1 activity on proliferation and cell death.
2. Additional gene expression analyses to compare the transcriptional states of *Lkb1*-restored and *Lkb1* tumors to that of normal alveolar type II epithelial cells.
3. Profiling of NKX2-1 and HMGA2 expression by immunohistochemistry across *Lkb1*-deficient tumors in the *Lkb1*^{XTR} model to capture differentiation state at varying time points after tumor initiation.
4. Several major modifications of the discussion section to better reflect the caveats of the *Lkb1*^{XTR} model and our functional studies.

Response to Reviewer #1

In this manuscript Murray et al develop a genetically engineered mouse model of Kras-Lkb1 mutant lung cancer in which they subsequently restore Lkb1 activity in established tumors. First, they demonstrate that restoration of Lkb1 over a 6 week period after establishing tumors for 6 weeks prevents further tumor growth. They then establish tumors for longer periods (18-22 weeks) and assess the more acute impact of Lkb1 restoration on tumor cell transcriptomic profiles after 2 weeks, identifying C/ebp transcription factor motifs and ATII gene signatures enriched in restored tumors in bulk analyses. They also perform single cell RNAseq which again shows enrichment for AT-like gene sets. To confirm a role for C/ebp in regulating this program, they also knockout C/ebp in Kras-Lkb1 tumors and also observe induction of ATII-like program. Finally they identify C/EBPalpha as a dominant factor co-operating in part with NKX2-1 to drive this program downstream of LKB1.

In general this is an extremely thorough, well done body of work. I have only a few minor suggestions, largely caveats of the work that should be mentioned in the discussion.

We thank this Reviewer for their critique of our work and noting its “extremely thorough, well-done” nature.

Q1. Fig 1 – while they observe tumor stasis, this does not differentiate between whether Lkb1 restoration causes cell cycle arrest/differentiation of all tumor cells, or whether it kills off a population of cells, selecting for those that end up with the ATII phenotype. Have they also stained for markers of apoptosis such as cleaved caspase 3, as this could support a selective process if positive? Unless they have strong evidence that this is not the case this caveat should be added to the discussion.

A1. This Reviewer raises an important question with respect to the nature of the response to *Lkb1* restoration in neoplastic cells. We describe the response to *Lkb1* restoration as a reprogramming event in which alveolar type II (ATII)-like identity is reinforced and coincides with proliferation arrest. However, this Reviewer has pointed out that, rather than driving transcriptional reprogramming, *Lkb1* restoration may instead impose a negative selection pressure against neoplastic subpopulations that lack features of ATII differentiation. In this scenario, the detection of ATII-like cells within restored tumors would result from the persistence of arrested/lowly cycling neoplastic cells that retain ATII-like features rather than direct reprogramming from a less-differentiated state. Several lines of evidence suggest that a model of reprogramming rather than selection drives our observations.

Response to Reviewers
NCOMMS-21-24931-T

To address this point, we have performed *in vitro* analyses to characterize the acute response to *Lkb1* restoration in terms of its impact on cell cycle and cell death in mouse lung cancer cell lines harboring the *Lkb1*^{XTR} allele. Within 96 hours of restoring *Lkb1*, we note a significant reduction in the fraction of cells in S phase (**new Extended Data Fig. 7D, E; see lines 163-167**). At the same time point, we did not observe a consistent increase in the fraction of apoptotic cells as a consequence of *Lkb1* restoration (**new Extended Data Fig. 7F, G; see lines 163-167**). These findings *in vitro* are consistent with our observations in the tumor setting in which we found a significant decrease in the number of proliferating cells after two weeks of initiating *Lkb1* restoration (**Fig. 2D, E and Extended Data Fig. 7A-C**). We also stained for cleaved caspase 3 and did not observe a significant change in the number of apoptotic cells between *Lkb1*-restored and non-restored tumors (**Fig. 2D, F**). Collectively, these observations favor a model in which *Lkb1* restoration directly drives the reprogramming of neoplastic cells towards a more differentiated, less proliferative state as opposed to promoting selection for more differentiated, less proliferative subpopulations.

Beyond examination of markers of proliferation and cell death, our single cell analyses also support a reprogramming model. Across *Lkb1*-restored and non-restored tumors we noted the presence of four major transcriptional states, two of which resemble ATI and ATII fates (**Fig. 4B, C**). We also identified a Krt8/Krt19-expressing subpopulation that resembles a transitional cell state that emerges during the regenerative response in the distal lung epithelium. Finally, we uncovered an indeterminate population that appears to be transcriptionally similar to the ATII-like state (**Fig. 4B, C**). Using trajectory inference approaches, we determined that this indeterminate subpopulation represents a transition state between the ATII and ATI fates (**Fig. 4D, E, Extended Data Fig. 15B**). Strikingly, we found that *Lkb1* restoration drives a shift in tumor composition from larger fraction of indeterminate cells in non-restored tumors to a large fraction of ATII-like cells in *Lkb1*-restored tumors (**Fig. 4F**). This finding suggested that *Lkb1* deficiency impairs complete ATII differentiation, thus trapping neoplastic cells in an indeterminate state (**Fig. 6H**). Together, these analyses further support a reprogramming event underlying the emergence of ATII identity within *Lkb1*-restored tumors.

Despite these findings, it remains possible that both differentiation and selection are occurring. Thus, we have now included new discussion within the main text to address the potential that an indirect, selection-driven mechanism could also contribute to the emergence of features of ATII differentiation within *Lkb1*-restored tumors (see **lines 543-559**).

Q2. Figs 2/3 – again this set of experiments is very elegant and convincing. However, parallel studies of restoring *Lkb1* *in vitro* in an established cell line in a time-controlled fashion would further help to address my question above, and whether the “acute” transcriptomic effects are

Response to Reviewers
NCOMMS-21-24931-T

direct, or a consequence of selection for a cell state that can tolerate *Lkb1* reconstitution. In their prior study (Murray *et al.*, Cancer Discov 2019) the authors examined gene sets in human *LKB1* reconstituted cell lines implicating SIK as a downstream activity. Have they similarly examined C/EBPalpha/ATII gene sets in this human cell line data (with the caveat that, depending on the time point after *LKB1* reconstitution, one may see different signatures).

A2. We have now extended our gene set enrichment analysis of C/EBP activity and alveolar epithelial cell identity to encompass *LKB1* rescue studies in human lung cancer cell lines (see Reviewer-only Table 1). Interestingly, we find that ATI signatures are enriched in *LKB1* rescue cells as compared to *LKB1*-deficient cells, whereas ATII signatures are enriched within *LKB1*-deficient cells in comparison to *LKB1* rescue cells. Similarly, we find that signatures of C/EBP-dependent genes are actually lower in the *LKB1* rescue state as compared to the *LKB1*-deficient setting. This contradicts with our findings in mouse lung tumors where we observe an increase in the expression of features of ATII identity (**Fig. 3D-H**). Primary ATII cells are known to rapidly differentiate into ATI cells within several days in standard two-dimensional culture and instead require more sophisticated culture systems, like air-liquid interface designs¹ and three-dimensional heterotypic organoids², to promote the persistence of ATII functions^{3, 4, 5, 6}. Thus, we suspect that the standard two-dimensional culture conditions employed in the studies that generated these previously published gene expression datasets may redirect the response to *LKB1* rescue, promoting the expression of features of ATI differentiation as opposed to those belonging to the ATII lineage. At present, genetic manipulation within organoids of lung epithelium is still in early stages, thus it is challenging to manipulate *LKB1* function within culture conditions that are actually conducive to the maintenance of ATII differentiation⁷.

Q3. The discussion feels a bit slanted against the role of canonical AMPK signaling and intent on proving the novel role for SIK kinases and the ATII differentiation program (model in 5H). However, I feel that it would be helpful to include discussion of the caveats above. Specifically, while everything may be direct, an alternate possibility is they are uncovering how KRAS-*LKB1* mutant cells successfully adapt to restoration of AMPK. For example, activation of YAP transcriptional programs can compensate for oncogenic KRAS suppression in an adaptive fashion, which is not direct.

A3. This Reviewer brings attention to the importance of highlighting the potential for an indirect, adaptive mechanism underlying the detection of features of ATII differentiation within *Lkb1*-restored tumors. It was not our intent to discount potential AMPK or other mechanisms. As noted in the main text, we observed that *Sik*-targeted tumors, like *Lkb1*-deficient tumors, exhibit lower expression of markers of ATII identity relative to lung tumors driven by oncogenic KRAS alone (**Extended Data Fig. 11G**). This suggests that

Response to Reviewers
NCOMMS-21-24931-T

the maintenance of features of ATII differentiation depends upon LKB1 and SIKs. At present, there are no publicly available datasets through which we can assess the role of AMPK in the maintenance of features of ATII differentiation. Nonetheless, AMPK may still play a part in maintaining ATII differentiation, particularly given that AMPK has been shown previously to be required for endodermal specification during embryoid body formation⁸. To address this point and make the discussion more balanced, we have modified the discussion to highlight the potential for an adaptive mechanism underlying the emergence of ATII differentiation as well as the potential for AMPK (or other factors) to be involved in this process (see lines 543-559).

Response to Reviewer #2

Murray et. al. demonstrate that restoration of the tumor suppressor LKB1 drives tumor cell growth arrest and transcriptional alveolar differentiation in lung adenocarcinoma (LUAD) cells. LKB1/STK11 is a frequently mutated tumor suppressor gene in treatment refractory human LUADs and its functions remains a topic of debate. Major strengths of the manuscript include the use of an elegant strategy (previously reported by the authors) to generate a genetically engineered mouse model (GEMM) whereby LKB1 expression can be restored in established tumors that initially arose in situ (in combination with Kras or Kras/p53 mutations). While loss of LKB1 in GEMM of LUAD has been previously reported, this study is the first to rigorously examine the consequences of LKB1 restoration (in a specifically controlled and timed manner) in vivo. The contribution of alveolar transcriptional programs to LUAD progression and their epigenetic regulation by lineage transcription factors (including C/EBPalpha and NK2X.1) is a known concept. However, the causal link between LKB1 loss of function and these lineage specific programs has not been clearly established. Therefore, on balance, the findings in Murray et al. are appropriate in terms of scope and novelty for Nature Communications.

We thank this Reviewer for their critical assessment of our work, as well as their appreciation of the elegance of this approach to study tumor suppressor function *in vivo* and the rigor with which we characterize the response to *Lkb1* restoration. We entirely agree that the key driving point of our work is the linkage of a critical tumor suppressor to lineage-specific transcription factor activity.

One important weakness is that the investigators do not provide any data regarding the grade and/or lineage marker status of their bulk tumors in vivo at the various time points of their experiments. It is particularly relevant to more clearly describe this when LKB1 restoration is initiated for several reasons. First, it is known that during LUAD progression (using similar Kras/p53 GEMM models) different lineage factors can be activated or suppressed as compensatory mechanisms of tumor suppression, and their expression or activity during tumor stasis or proliferation may be stage specific. Second, these GEMMs are likely to progress and transdifferentiate in a heterogeneous manner. Third, assuming the feasibility of re-activating tumor suppressor pathways in human NSCLC (as suggested by the authors) knowing what grade or human equivalent stage of LUAD are most susceptible to LKB1 restoration would help the readers interpret the potential impact of the findings. It is also possible that some genetic or stage specific context will not be responsive to LKB1 restoration alone when a so -called intermediate AT2/AT1 like state (the presumed target of restored LKB1 activity) has already completely eroded (see additional comments regarding p53 below).

We thank this Reviewer for their emphasis on the potential for context-specificity in terms of the response to *Lkb1* restoration. We entirely agree that, at least at later stages of

Response to Reviewers
NCOMMS-21-24931-T

tumor development, differentiation state heterogeneity among lung tumors in the $Kras^{LSL-G12D/+};p53^{lox/lox}$ model is well studied^{9, 10}. The $Kras^{LSL-G12D/+};Lkb1^{lox/lox}$ (*KL*) model has also been reported to yield tumors of diverse differentiation^{11, 12, 13, 14}. However, the dynamics with which tumors in the *KL* model diverge from the standard adenocarcinoma trajectory are not yet appreciated.

As the most dramatic example, tumors in both the *KP* and *KL* models often progress to a very poorly differentiated states that lack the expression of the key lung epithelial transcription factor TTF1/NKX2-1. To assess the differentiation state of lung tumors at the various timepoints in which we initiated *Lkb1* restoration, we have performed immunohistochemical staining on sections from tumor-bearing lungs from $Kras^{LSL-G12D/+};Lkb1^{XTR/XTR}$ mice for NKX2-1, a marker of adenocarcinoma differentiation, and HMGA2, a marker of poorly differentiated regions (**new Extended Data Fig. 2L, M; see lines 99-102**)¹⁵. Across five timepoints (spanning 12-28 weeks post tumor initiation), we observed limited evidence of non-adenocarcinoma differentiation among *Lkb1*-deficient tumors developing within $Kras^{LSL-G12D/+};Lkb1^{XTR/XTR}$ mice. This is consistent with prior histological characterization of our $Kras^{LSL-G12D/+};Lkb1^{XTR/XTR}$ model in which we noted predominantly adenocarcinoma differentiation accompanied by rare mucinous adenocarcinoma¹³. Notably, consistent with our previous study in $Kras^{LSL-G12D/+};Lkb1^{lox/lox}$ mice, we did not observe the development of squamous or adenosquamous tumors¹³. From this analysis, we conclude that response to *Lkb1* restoration that we characterize here is largely specific to tumors of adenocarcinoma differentiation.

We agree with this Reviewer that there may be more nuance to the response to tumor suppressor gene restoration. Stage-specificity could be important (as described previously for p53 restoration) while effects across different genetic background is likely critical but has not been addressed in any tumor suppressor restoration studies. To address this point, we have updated the discussion to comment on the potential for the response to *Lkb1* restoration (and tumor suppressor function in general) to be influenced by tumor stage or genetic context (**see lines 493-503**).

Specific comments, questions, and suggestions are as follows:

Q1. Fig 4 and S3: It was not always clear in the legends how the sampling was done. For example, how many tissue sections were used to assess tumor area and how many mice were used to quantify tumor size distributions?

A1. We thank this Reviewer for bringing to our attention the need for greater clarity regarding the manner in which we report metrics of tumor burden. For all experiments

Response to Reviewers
NCOMMS-21-24931-T

including measures of tumor area and tumor size, a single section of tumor-bearing lung from each mouse was analyzed. We have updated figure legends to indicate the number of sections analyzed per mouse, as well as the number of mice analyzed for each experiment.

Q2. Figure S5: It is somewhat misleading to state that the effect of LKB1 restoration is qualitatively independent of p53 without commenting on the quantitative differences; in the absence of p53, the effects on tumor stasis are clearly more muted. This is also reflected in the gene expression analysis (Fig. S11). Is this because the grade of the tumors in p53 null animals are higher (when LKB1 is initially restored) or is there a context dependent molecular explanation for this, given the reported reciprocal relationship between LKB1 and p53 (in LUAD)? A more accurate description and discussion of these results is warranted.

A2. We agree with this Reviewer that we initially did not fully acknowledge the less-pronounced impact of *Lkb1* restoration in the p53-deficient setting as compared to p53 wild-type. We have now modified the results section of the main text to reflect this discrepancy and provide additional comments in the discussion (see lines 139-144, 496-499).

Q3. Fig S7 and S8: Some of the major data should be included as main figures, (journal policies allowing). Statistical significance seems to be missing for S8 in particular.

A3. We agree that the significance of the findings encompassed within **Extended Data Fig. 7** and **Extended Data Fig. 8** warrants their prioritization as a main figure. We have now generated an additional main figure that centers primarily on the longitudinal μ CT and ^{18}F -FDG PET/CT imaging data (new Fig. 2). To address the request for the testing of statistical significance in the context of the ^{18}F -FDG PET/CT imaging data, we have generated a new summary figure panel (new Fig. 2H) that indicates that a significant increase in max ^{18}F -FDG uptake relative to pre-treatment levels was detectable at six weeks post treatment initiation.

Q4. While much of the supplemental transcriptomic/proteomic analysis are rigorous, this data is descriptive and, in some cases, somewhat redundant (e.g., S9 and S10, S17); It would be useful if some of this processed data could be summarized and then presented fully as supplemental tables for exploration by the readers.

A4. We thank the Reviewer for this helpful suggestion regarding the presentation of our transcriptomic and proteomic data. We now include aggregated counts and intensity matrices corresponding to our bulk RNA-seq and shotgun proteomics experiments as supplementary materials (see Supplementary Tables 1-11).

Q5. Fig S15 legend and related text: It is recommended that the authors clarify in the text that the lineage specific proliferation status is inferred by analyzing cell cycle gene markers from single cell RNA-seq data. It also seems highly likely that LKB1 suppresses proliferation in lineages that are not AT2 like. The conclusion in the manuscript could be more justified if functional confirmation in tumor tissue was provided; otherwise, the stated conclusions should be more aligned with the nuanced and indirect data that is presented.

A5. We thank this Reviewer for highlighting the need for greater clarity with respect to the examination of proliferating and non-proliferating fractions of neoplastic cells. We did not directly ascertain the proliferative potential for each transcriptional state identified across *Lkb1*-restored and non-restored tumors. We instead compared the relative abundance of each transcriptional state between proliferative and non-proliferative fractions as defined by the expression of genes relating to cell cycle progression (**Extended Data Fig. 15F**). In doing so, we noted that ATII-like state was significantly underrepresented within the proliferative fraction as compared to the non-proliferative fraction, suggesting that the ATII-like state is less proliferative. Regarding the remaining cell states, we did not detect a significant difference in terms of representation between proliferative and non-proliferative subpopulations (**Extended Data Fig. 15F**). However, we do not exclude the possibility that *Lkb1* restoration suppresses proliferation within non-ATII-like states (see new Extended Data Fig. 15G; lines 344-347). To provide further clarification, we include additional emphasis in the results section that the conclusions concerning proliferation were derived from single cell RNA-seq analysis (see lines 337-347).

Response to Reviewer #3

LKB1 is among the most frequently altered tumor suppressors in lung adenocarcinoma. In this manuscript, Murray et al. develop an elegant system to test whether restoration of Lkb1 expression would limit tumor progression in a KrasG12D/Lkb1-LOF mouse lung tumor model. They validate a novel mouse model (Lkb1XTR), in which Cre-mediated Lkb1 loss of function can be complemented by tamoxifen-regulated FLPo-ER mediated restoration of Lkb1 expression to examine roles for Lkb1 loss in tumor progression. Using this model, the authors demonstrate that restoration of Lkb1 expression does indeed constrain tumorigenesis driven by the combination of Kras/Lkb1 loss of function. Further, the authors identify targets of C/EBP transcription factors as key downstream targets of Lkb1 that regulate tumor progression. The authors also demonstrate that Lkb1 restoration drives transcriptional programs relating to alveolar type 2 epithelial cell function, suggesting that roles for Lkb1 in the regulation of alveolar epithelial cell differentiation plays a role in tumor suppression. Overall, experiments are well designed, data of high quality, and are generally supportive of conclusions.

We thank this Reviewer for their critique of our work and appreciation for the elegance of this *in vivo* system to investigate *Lkb1* function.

Concerns:

Q1. Multiple publications have shown that inactivation of Lkb1 in Kras driven mouse lung tumor leads to adenosquamous transition, and emergence of diverse histological subtypes. However, this is not seen in this study. The authors should discuss this issue further to help reconcile their findings with the existing literature.

A1. Squamous cell carcinoma (SCC) has been reported previously in multiple studies involving *Kras*^{LSL-G12D/+};*Lkb1*^{fllox/fllox} (*KL*) mice. We have compiled a comprehensive summary of these previous reports to illustrate the variable incidence of SCC in *KL* models and describe the technical variations across these studies (see Reviewer-only Table 2). SCC tumors in these models display unique histological features and express canonical SCC markers, such as p63 and CK5. We have shown previously that SCC does not present within our *KL* model (Murray et al. 2019 Cancer Discovery: Extended Data Fig. 8C, D). Specifically, we analyzed nearly 200 tumors for p63 and CK5 expression as well as had our H&E-stained sections reviewed by a board-certified lung cancer pathologist. From this data we concluded that SCC does not emerge in our *KL* model. Thus, it is not unexpected that we did not identify SCC within the *Kras*^{LSL-G12D/+};*Lkb1*^{XTR/XTR} model reported here.

For reasons that likely stem from varying tropism between adenoviral and lentiviral vectors, the presentation of SCC has typically been noted in *KL* mice in the context of

Response to Reviewers
NCOMMS-21-24931-T

Ad-Cre-driven tumor initiation^{12, 14, 16, 17, 18, 19, 20, 21, 22, 23} and less so in the setting of Lenti-Cre transduction^{11, 13}. The lack of SCC formation in the *KL* model following Lenti-Cre-initiated tumorigenesis has been reported previously by Adam Marcus' group¹¹. We have discussed this observation with him in the past, and he confirmed that, across hundreds of *KL* mice with Lenti-Cre-initiated tumors, they have never observed evidence of SCC formation. Thus, given that our *Kras*^{LSL-G12D/+}; *Lkb1*^{XTR/XTR} model employs Lenti-Cre, it is not unexpected that these mice would not develop SCC.

However, it is unlikely that the discrepancy in SCC incidence is simply explained by varying viral tropism. Hongbin Ji's group has extensively employed the *KL* model to make critical discoveries concerning lung SCC, and they have reported SCC development in the context of Lenti-Cre-driven tumor initiation^{12, 16, 17, 19}. Thus, it is plausible that other differences between the mouse models used in different studies could influence SCC incidence. For instance, mouse strain could be one contributing factor. Our *Kras*^{LSL-G12D/+}; *Lkb1*^{XTR/XTR} colony is a mix of 129 and BL6, while other investigators have used *KL* on FVB/N or other combinations of strains).

Finally, it is important to note that mutations in *KRAS* and *LKBI* mutations are very infrequent in human lung SCC. According to cBioPortal, only four out of 469 tumors (~0.9%) in the TCGA SCC cohort have mutations in codons 12, 13, or 61 of *KRAS* and eight out of 469 (~1.7%) harbor mutations or putative homozygous copy number loss in *LKBI*²⁴. Thus, while the SCC described by others in *KL* mice have led to great insights, the lack of SCC tumors in our model is largely consistent with human genomic data and previous Lenti-Cre-initiated mouse models.

Q2. A caveat with the LkbXTR model is that "restoration" of Lkb1 occurs not only within tumors but presumably increased Lkb1 occurs in all cells undergoing tamoxifen regulated FLPo mediated recombination. Does this lead to changes in the molecular phenotype of non-tumorigenic cells? In general, additional discussion should be included that identifies caveats with the model that impact interpretation of data.

A2. We thank this Reviewer for bringing this item to our attention. The XTR system was designed to disrupt gene function in a Cre-dependent manner, thus only those cells in which Cre has been expressed, either through delivery on a viral vector or from a germline driver allele, will be rendered deficient for a gene of interest. Subsequent FLPo-ER^{T2}-mediated deletion of the XTR gene trap cassette (in both its expressed and trapped conformations) enables the restoration of endogenous levels of expression of a gene of interest within cells that were previously deficient (**Fig. 1A and Extended Data Fig. 1C**). Thus, restored expression of a gene of interest is restricted to cells that have been

Response to Reviewers
NCOMMS-21-24931-T

previously exposed to Cre activity, as cells harboring the XTR allele in the expressed conformation would not have been previously rendered deficient for the gene of interest.

In the case of *Lkb1*^{XTR} allele, the XTR gene trap, in its expressed conformation, impairs *Lkb1* expression at the mRNA and protein levels, thus rendering mice globally hypomorphic for *Lkb1* (**Extended Data Fig. 2A, B**). Consequently, the induction of FLPo-ER^{T2}-mediated deletion of the XTR gene trap cassette not only restores the expression of *Lkb1* within Cre-expressing neoplastic cells but also results in elevated expression of *Lkb1* in cells outside of the neoplastic compartment (**see new Extended Data Fig. 13E; lines 296-297**). Therefore, at least in the autochthonous setting, one cannot entirely exclude the potential impact on tumor growth dynamics of increasing *Lkb1* expression globally from hypomorphic levels upon induction of FLPo-ER^{T2} activity.

To circumvent this limitation, we investigated the effect of *Lkb1* restoration in an allograft setting, thus allowing us to restrict changes in *Lkb1* expression to neoplastic cells as opposed to other neighboring non-neoplastic cells (**Extended Data Fig. 4**). In this setting, we found that *Lkb1* restoration halted the growth of established tumors and impaired the ability of neoplastic cells to engraft within recipient lungs, thus indicating that the deleterious effects of *Lkb1* restoration can be largely attributed to increased expression of *Lkb1* within the neoplastic cells alone. We have modified the legend for **Extended Data Fig. 4A** (**see updated legend for Extended Data Fig. 4A**) to provide further emphasis on the importance of investigating the effects of *Lkb1* restoration in a context in which the manipulation of *Lkb1* expression is restricted to cancer cells alone.

Regarding the question as to whether there is any evidence of molecular changes within non-neoplastic cells in response to the global increase in *Lkb1* expression, we did not note any significant changes in gene expression outside of the neoplastic compartment as assessed by single cell RNA-seq (**Extended Data Fig. 13G**). However, we did identify a small subset of genes that were differentially expressed within the myeloid and endothelial compartments when comparing cells from non-restored and restored tumors. Among T cells, we noted significantly higher expression of *Ccl5* and significantly lower expression of members of the *Hist1* cluster, including *Hist1h4d*, *Hist1h4e*, and *Hist1h2ab*, in the restored setting as compared to non-restored. Within the endothelial cells, we noted significantly elevated expression of interferon-induced genes, *Ifi2712a* and *Iigp1* in response to *Lkb1* restoration (**see source data for Extended Data Fig. 13G**). However, we would like to point out that these gene expression changes in the non-neoplastic cells may not be due to an intrinsic increase in *Lkb1* expression but could represent secondary changes induced by signals emanating from neoplastic cells in which *Lkb1* has been restored.

Q3. Authors claim that restoration of *Lkb1* drives transcriptional programs relating to alveolar type 2 cell function. However, data in Figure 2C suggest that “*Lkb1*-restored” tumor cells acquire a molecular phenotype more like those of *Kras*-induced tumors. Direct comparisons to wildtype AT2 cells are not made. These comparisons would be helpful in assessing changes in molecular phenotype of *Lkb1*-sufficient, -deficient (*KT*) and “restored” tumor cells. These comparisons would also provide insights into the relationship between observations in this study with other related reports. Additional discussion would be helpful.

A3. We agree with this Reviewer that a direct comparison to wild-type ATII cells would be helpful in terms of determining where the transcriptional identity of *Lkb1*-restored tumors lies on the spectrum between wild-type ATII cells and oncogenic *KRAS*-expressing neoplastic cells. We do not have an overlapping RNA-seq experiment that includes purified mouse ATII cells to enable direct integration with our existing RNA-seq experiment.

To circumvent this inability to directly integrate independent bulk RNA-seq data sets, we have performed additional GSEA analyses using signatures of ATII identity derived from mouse ATII single cell RNA-seq data^{25, 26, 27, 28, 29, 30, 31, 32}. Specifically, we have examined the enrichment of signatures of ATII identity within *Lkb1*-restored tumors as compared to *Lkb1* wild-type (*KT*) tumors. Interestingly, there was no consistent enrichment of ATII signatures in either *Lkb1*-restored or *KT* tumors, though a single ATII signature was significantly enriched and several others trended towards enrichment in the *KT* setting (see new Extended Data Fig. 10G). This indicates that ATII identity is likely not the primary determinant underlying the difference in transcriptional identity between *Lkb1*-restored and *KT* tumors that is apparent in Fig. 2C (see lines 246-251).

To follow up on the distinction between *Lkb1*-restored tumors and *KT* tumors, we performed an additional round of GSEA using gene signatures from the GO Biological Process module. This analysis revealed that relative to *KT* tumors, the *LKB1*-restored state was significantly enriched with several signatures relating to morphogenesis and tissue development (see new Extended Data Fig. 10H; see lines 246-251).

Q4. In discussion, authors state that the inability of *LKB1* re-expression to “recover the innate immune components due to stable epigenetic repression ...” contrasts with their finding that *Lkb1* restoration suppresses tumor progression. How do the authors reconcile these observations – this should be discussed?

A4. We thank this Reviewer for their comment. In referencing the recent observations reported by Kitajima *et al.*, our aim was to highlight an instance in which *LKB1* rescue

Response to Reviewers
NCOMMS-21-24931-T

did not result in complete recovery of molecular features that function within *LKB1* wild-type cells. Rather than focus specifically on the expression of STING, our intent was to underscore the possibility that restoring *Lkb1* activity within our mouse model could have failed to elicit any response due to tumors shedding their dependency on *Lkb1* deficiency over the course of progression. We have now reworded this sentence to provide further clarity (see lines 485-492).

This Reviewer has asked why we observe tumor stasis in response to *Lkb1* restoration in mouse lung tumors, while, in some human lung cancer cell lines, LKB1 rescue results in incomplete reversion of the gene expression changes that incur from *LKB1* inactivation. It may be that the reconstitution of LKB1 activity is capable of re-engaging cancer cell intrinsic mechanisms of tumor suppression, like proliferative control, whereas subsets of genes that have undergone stable epigenetic silencing, such as the case of STING, may no longer be responsive to LKB1 activity. In other words, epigenetic changes stemming from LKB1 loss could result in selective inhibition of a subset of LKB1-dependent processes, while others remain intact.

Q5. The concluding sentence starting on line 529 addresses *Lkb1*-dependent changes in the proliferative activity of tumors. This seems somewhat disconnected from the remainder of the paragraph which focuses on the diversity of histological subtypes and how this may be mediated by altered expression of *Nkx2.1* and downstream target genes. Perhaps this paragraph should be re-focused for clarity.

A5. We thank this Reviewer for their comment. The paragraph under concern centers on tumor differentiation, thus we have dropped our final remark regarding proliferation from this paragraph, such that the concluding sentence aligns with the body of the paragraph (see lines 521-522).

Q6. Figure 3F and S13. Representation of cell types within scRNA-Seq data can not be directly equated with their abundance within tumors.

A6. We acknowledge that differential abundance analysis can be challenging when it involves non-discrete cell states^{33, 34, 35, 36}. However, comparisons of relative abundance of cell states within tissues across conditions or time has become an increasingly common practice in single-cell analysis^{37, 38, 39, 40, 41, 42, 43}. In fact, there have been two recent studies that center on the use of single cell profiling to capture changes in cellular composition of the neoplastic compartment throughout tumor progression in the *Kras*^{LSL-G12D/+}; *p53*^{flox/flox} lung cancer mouse model^{9, 10}. We show here that the ATII and indeterminate identities are distinguished by the activity of C/EBP-dependent genes, thus we are confident that these are indeed distinct cell states that are quantifiable in terms of

Response to Reviewers
NCOMMS-21-24931-T

their representation across *Lkb1*-restored and non-restored tumors (**Fig. 6B**). Furthermore, our bulk RNA-seq analysis suggest that the representation of the ATII state is greater within *Lkb1*-restored tumors as compared to non-restored (**Fig. 3E, F**)

Q7. In Supplemental Figure S4, the authors transplant tumor cells from KT;Lkb1XTR/XTR;FLPo-ERT2 to NSG mice, followed by treatment with either vehicle or tamoxifen. A significant reduction of tumorigenesis was observed among Lkb1-restored compared to Lkb1-LOF tumors. Have the authors made comparisons with KT transplantation to determine whether Lkb1 restoration is comparable to KT?

A7. We thank this Reviewer for their suggestion. We agree that an *Lkb1* wild-type control would have been informative in that it would allow for comparison with the *Lkb1*-restored cohort to assess the magnitude of the deleterious effects of *Lkb1* restoration. At the time of this experiment, we did not have sufficiently large *KT* tumors available for transplant. However, despite this deficiency, our conclusion that *Lkb1* restoration is deleterious to tumor engraftment and outgrowth remains supported by the existing data (**Extended Data Fig. 4**).

Q8. In Supplemental Figure S5, a KPT alone control will help with data interpretation.

A8. We agree that an *Lkb1* wild-type control would have been helpful for examining the effects of *Lkb1* restoration in the p53-deficient setting. Unfortunately, we did not have *KPT* mice available at the time that this experiment was conducted. Nonetheless, the data support our conclusion that *Lkb1* restoration lessens lung tumor burden in the p53-deficient context (**Extended Data Fig. 5**).

Q9. Typographical error in Supplemental Figure S11D: label of lower panel should be Higher in Non-Restored tumors.

A9. We thank this Reviewer for bringing this item to our attention. The apparent typographical error stemmed from the truncation of the figure panel image upon import into Microsoft Word. We have now adjusted the image margins to correct this error (see Extended Data Fig. 11D).

REFERENCES

1. Dobbs LG, Pian MS, Maglio M, Dumars S, Allen L. Maintenance of the differentiated type II cell phenotype by culture with an apical air surface. *Am J Physiol* **273**, L347-354 (1997).

Response to Reviewers

NCOMMS-21-24931-T

2. Barkauskas CE, *et al.* Type 2 alveolar cells are stem cells in adult lung. *J Clin Invest* **123**, 3025-3036 (2013).
3. Barkauskas CE, Chung MI, Fioret B, Gao X, Katsura H, Hogan BL. Lung organoids: Current uses and future promise. *Development (Cambridge, England)* **144**, 986-997 (2017).
4. Chen Q, *et al.* Angiocrine sphingosine-1-phosphate activation of S1PR2-YAP signaling axis in alveolar type II Cells Is essential for lung repair. *Cell Rep* **31**, 107828 (2020).
5. Dobbs LG. Isolation and culture of alveolar type II cells. *Am J Physiol* **258**, L134-147 (1990).
6. Hogan B, Tata PR. Cellular organization and biology of the respiratory system. *Nat Cell Biol*, (2019).
7. van der Vaart J, Clevers H. Airway organoids as models of human disease. *J Intern Med* **289**, 604-613 (2021).
8. Young NP, *et al.* AMPK governs lineage specification through Tfeb-dependent regulation of lysosomes. *Genes Dev* **30**, 535-552 (2016).
9. LaFave LM, *et al.* Epigenomic state transitions characterize tumor progression in mouse lung adenocarcinoma. *Cancer cell* **38**, 212-228.e213 (2020).
10. Marjanovic ND, *et al.* Emergence of a high-plasticity cell state during lung cancer evolution. *Cancer cell* **38**, 229-246.e213 (2020).
11. Gilbert-Ross M, *et al.* Targeting adhesion signaling in *KRAS*, *LKB1* mutant lung adenocarcinoma. *JCI insight* **2**, e90487 (2017).
12. Ji H, *et al.* LKB1 modulates lung cancer differentiation and metastasis. *Nature* **448**, 807-810 (2007).
13. Murray CW, *et al.* An LKB1-SIK axis suppresses lung tumor growth and controls differentiation. *Cancer Discov* **9**, 1590-1605 (2019).
14. Nagaraj AS, *et al.* Cell of origin links histotype spectrum to immune microenvironment diversity in non-small-cell lung cancer driven by mutant *Kras* and loss of *Lkb1*. *Cell Rep* **18**, 673-684 (2017).
15. Winslow MM, *et al.* Suppression of lung adenocarcinoma progression by Nkx2-1. *Nature* **473**, 101-104 (2011).

Response to Reviewers

NCOMMS-21-24931-T

16. Gao Y, *et al.* YAP inhibits squamous transdifferentiation of *Lkb1*-deficient lung adenocarcinoma through ZEB2-dependent DNp63 repression. *Nat Commun* **5**, 4629 (2014).
17. Han X, *et al.* Transdifferentiation of lung adenocarcinoma in mice with *Lkb1* deficiency to squamous cell carcinoma. *Nat Commun* **5**, 3261 (2014).
18. Hsieh MH, *et al.* p63 and SOX2 dictate glucose reliance and metabolic vulnerabilities in squamous cell carcinomas. *Cell Rep* **28**, 1860-1878.e1869 (2019).
19. Li F, *et al.* *LKB1* inactivation elicits a redox imbalance to modulate non-small cell lung cancer plasticity and therapeutic response. *Cancer cell* **27**, 698-711 (2015).
20. Momcilovic M, *et al.* Heightening energetic stress selectively targets *LKB1*-deficient non-small cell lung cancers. *Cancer Res* **75**, 4910-4922 (2015).
21. Singh A, *et al.* NRF2 activation promotes aggressive lung cancer and associates with poor clinical outcomes. *Clin Cancer Res* **27**, 877-888 (2021).
22. Tong X, *et al.* Nanog maintains stemness of *Lkb1*-deficient lung adenocarcinoma and prevents gastric differentiation. *EMBO Mol Med* **13**, e12627 (2021).
23. Zhang H, *et al.* *Lkb1* inactivation drives lung cancer lineage switching governed by Polycomb Repressive Complex 2. *Nat Commun* **8**, 14922 (2017).
24. Cerami E, *et al.* The cBio Cancer Genomics Portal: An open platform for exploring multidimensional cancer genomics data. *Cancer Discov* **2**, 401 (2012).
25. Angelidis I, *et al.* An atlas of the aging lung mapped by single cell transcriptomics and deep tissue proteomics. *Nat Commun* **10**, 963 (2019).
26. Guo M, *et al.* Single cell RNA analysis identifies cellular heterogeneity and adaptive responses of the lung at birth. *Nat Commun* **10**, 37 (2019).
27. Han X, *et al.* Mapping the Mouse Cell Atlas by Microwell-Seq. *Cell* **172**, 1091-1107.e1017 (2018).
28. Little DR, *et al.* Transcriptional control of lung alveolar type 1 cell development and maintenance by NK homeobox 2-1. *Proc Natl Acad Sci U S A* **116**, 20545-20555 (2019).
29. Schaum N, *et al.* Single-cell transcriptomics of 20 mouse organs creates a *Tabula Muris*. *Nature* **562**, 367-372 (2018).
30. Schaum N, *et al.* Ageing hallmarks exhibit organ-specific temporal signatures. *Nature* **583**, 596-602 (2020).

Response to Reviewers

NCOMMS-21-24931-T

31. Strunz M, *et al.* Alveolar regeneration through a Krt8+ transitional stem cell state that persists in human lung fibrosis. *Nat Commun* **11**, 3559 (2020).
32. Treutlein B, *et al.* Reconstructing lineage hierarchies of the distal lung epithelium using single-cell RNA-seq. *Nature* **509**, 371-375 (2014).
33. Burkhardt DB, *et al.* Quantifying the effect of experimental perturbations at single-cell resolution. *Nat Biotechnol* **39**, 619-629 (2021).
34. Büttner M, Ostner J, Müller CL, Theis FJ, Schubert B. scCODA: A Bayesian model for compositional single-cell data analysis. *bioRxiv*, 2020.2012.2014.422688 (2020).
35. Dann E, Henderson NC, Teichmann SA, Morgan MD, Marioni JC. Differential abundance testing on single-cell data using k-nearest neighbor graphs. *Nat Biotechnol*, (2021).
36. Zhao J, *et al.* Detection of differentially abundant cell subpopulations in scRNA-seq data. *Proc Natl Acad Sci U S A* **118**, e2100293118 (2021).
37. Chua RL, *et al.* COVID-19 severity correlates with airway epithelium-immune cell interactions identified by single-cell analysis. *Nat Biotechnol* **38**, 970-979 (2020).
38. Kinchen J, *et al.* Structural remodeling of the human colonic mesenchyme in inflammatory bowel disease. *Cell* **175**, 372-386.e317 (2018).
39. Liao M, *et al.* Single-cell landscape of bronchoalveolar immune cells in patients with COVID-19. *Nat Med* **26**, 842-844 (2020).
40. Pijuan-Sala B, *et al.* A single-cell molecular map of mouse gastrulation and early organogenesis. *Nature* **566**, 490-495 (2019).
41. Ramachandran P, *et al.* Resolving the fibrotic niche of human liver cirrhosis at single-cell level. *Nature* **575**, 512-518 (2019).
42. Sade-Feldman M, *et al.* Defining T cell states associated with response to checkpoint immunotherapy in melanoma. *Cell* **175**, 998-1013.e1020 (2018).
43. Smillie CS, *et al.* Intra- and inter-cellular rewiring of the human colon during ulcerative colitis. *Cell* **178**, 714-730.e722 (2019).

REVIEWERS' COMMENTS

Reviewer #1 (Remarks to the Author):

The authors have satisfactorily addressed my concerns

Reviewer #2 (Remarks to the Author):

The manuscript has been reviewed to include: new in vivo characterization of tumor differentiation and proliferation in the GEMM at different timepoints, new complementary in vitro cellular data, more nuanced and informative description of the data in Results and Discussion sections, and re-organization of the figures. The manuscript is well-balanced and rigorous. I have no remaining comments to add.

Reviewer #3 (Remarks to the Author):

Authors have comprehensively responded to reviewer concerns.